# Emergence of the cortical encoding of phonetic features in the first year of life

Giovanni M. Di Liberto [1,2,3,7] ✉, Adam Attaheri [3,7], Giorgia Cantisani[1,4], Richard B. Reilly [2,5,6], Áine Ní Choisdealbha [3], Sinead Rocha[3], Perrine Brusini[3] & Usha Goswami [3]

Even prior to producing their first words, infants are developing a sophisticated speech processing system, with robust word recognition present by 4–6 months of age. These emergent linguistic skills, observed with behavioural investigations, are likely to rely on increasingly sophisticated neural underpinnings. The infant brain is known to robustly track the speech envelope, however previous cortical tracking studies were unable to demonstrate the presence of phonetic feature encoding. Here we utilise temporal response functions computed from electrophysiological responses to nursery rhymes to investigate the cortical encoding of phonetic features in a longitudinal cohort of infants when aged 4, 7 and 11 months, as well as adults. The analyses reveal an increasingly detailed and acoustically invariant phonetic encoding emerging over the first year of life, providing neurophysiological evidence that the pre-verbal human cortex learns phonetic categories. By contrast, we found no credible evidence for age-related increases in cortical tracking of the acoustic spectrogram.

The human ability to understand speech relies on a complex neural system, whose foundations develop over the first few years of life. A wealth of evidence on the developmental progression of speech processing is available from infant behavioural studies, including with neonates, augmented by studies of speech production from around the second year of life[1–3]. Experiments using behavioural measures enable the assessment of valuable factors such as the familiarity of a particular speaker, the phonetic features that can be discriminated, and sensitivity to native versus non-native speech contrasts, and have been assumed to provide a timeline for 'cracking the speech code' in the first year of life[1]. Yet our understanding of infant speech processing in the first year of life is largely dependent on tasks relying on simple behaviours (e.g., head turn preference procedures). Ideally, this understanding should be complemented by studies determining the

neural encoding of phonological information across the first year of life, using natural listening tasks and continuous speech. Previously, methodological limitations have forced neural studies with infants to rely on discrete rather than continuous stimuli, and electrical measurements of neural activity have relied on evoked potentials[3,4].

Continuous speech listening has been difficult to study neurally because the most widely used neural encoding paradigms with infants have measured the mismatch negativity (MMN), or mismatch response (MMR), which requires discrete stimuli. The MMR is a neurophysiological signature of automatic change detection[3,5,6] typically used to measure the ability to discriminate phonetic categories, for example by measuring neural changes to 'ba' vs 'pa'. Further, previous EEG studies showed that such mismatch responses in infants can sometimes be positive[7], causing inconsistencies that can complicate or limit

[1]ADAPT Centre, School of Computer Science and Statistics, Trinity College, The University of Dublin, Dublin, Ireland. [2]Trinity College Institute of Neuroscience, Trinity College, The University of Dublin, Dublin, Ireland. [3]Centre for Neuroscience in Education, Department of Psychology, University of Cambridge, Cambridge, United Kingdom. [4]Laboratoire des Systémes Perceptifs, Département d'études Cognitives, École normale supérieure, PSL University, CNRS, 75005 Paris, France. [5]School of Engineering, Trinity Centre for Biomedical Engineering, Trinity College, The University of Dublin., Dublin, Ireland. [6]School of Medicine, Trinity College, The University of Dublin, Dublin, Ireland. [7]These authors contributed equally: Giovanni M. Di Liberto, Adam Attaheri. ✉e-mail: gd467@cam.ac.uk; diliberg@tcd.ie

their use in infants. Accordingly, measuring MMRs typically constrains researchers to using simple listening scenarios (e.g., sequences of isolated syllables) and to focus on only a few selected phonetic contrasts. Recently there have been notable advances, such as measuring the EEG encoding of multiple vowel sounds at the same time[8]. This cross-sectional study went from infancy to adulthood and found compelling evidence regarding the formation of the perceptual vowel space in early development[8]. However, even in this case, the stimuli were far removed from natural speech, as they consisted of continuous sequences of vowels (with no consonants), excluding all other phonological properties of speech, from consonants to prosody. This leaves us with a key open question regarding the neurophysiology of early speech processing of phonetic categories: when do infants reliably process phonological units such as /b/ versus /p/ in continuous natural speech? This study aims to shed light on that question by measuring if these speech sounds are encoded as categorical units in the infant brain, and by determining when exactly that acoustically invariant encoding emerges across the first year of life.

This study investigates acoustic and acoustically invariant encoding of speech sounds across the first year of life by measuring the neural tracking of speech. The neural tracking of (or neural entrainment in the broad sense[9]) to stimulus features such as the acoustic envelope[10–13] offers a direct window into the neural processing of speech during natural listening without imposing any particular task other than listening. In recent years, neural tracking measures have played a growing role in the study of speech comprehension and auditory processing in general. Previous studies with adults have assessed the neural tracking of the acoustic envelope[10–13], which is an important property of speech that co-varies with a number of key properties of interest (e.g., syllable stress patterns, syllables, phonemes). Neural tracking of the speech envelope (or envelope tracking) was shown to reflect both bottom-up and top-down cortical processes in adult listeners, encompassing fundamental functions such as selective attention[13–15], working memory processing load[16], and prediction[17,18]. While robust envelope tracking has also been demonstrated in infants[19–24], the previous envelope measures only revealed some of the cortical mechanisms underlying speech perception. Recent work with adults and children has demonstrated that neural tracking measurements can also be used to isolate the cortical encoding of targeted speech properties of interest, from phonetic features[25,26] and phonotactics[27,28] to semantic dissimilarity[29,30] and surprise[28,30,31]. Phonetic encoding was measured in neural tracking studies from different research teams[25,28,32–34] and was shown to correlate with phonemic awareness skills in school-aged children between 6 and 12 years of age[35] and with second language proficiency in adults[36]. Recent work applying the same methodology to intracranial electroencephalography recordings was also able to pinpoint the cortical origins of phonetic feature encoding[37].

Here, we use neural tracking measurements to assess the neural encoding of the full phonetic feature inventory of continuous speech, applying non-invasive electroencephalography (EEG) in an ecologically valid paradigm of singing to an infant. EEG signals were recorded as infants listened to 18 nursery rhymes (vocals only with no instruments involved) presented via video recordings of a native English speaker. EEG recordings were carried out at 4, 7 and 11 months of age. We then measured how the infant brain encodes acoustic and phonetic information by means of the multivariate Temporal Response Function analysis (TRF), a neurophysiology framework enabling the measurement of neural tracking by relating neural signals with multiple features of a continuous sensory stimulus[38,39]. TRF analyses were also carried out on recordings from adult participants listening to the same audio-visual nursery rhyme stimuli. We targeted one key aspect for speech processing: the neural encoding of phonetic feature categories. We do not assume here that encoding phonetic features equates to encoding phonemes, as there is a large psychoacoustic and developmental literature showing that phonemes are only represented by literate brains[40,41]. Our core hypothesis was that phonetic feature encoding (invariant to acoustic changes) would emerge in the neural responses to nursery rhymes during the first year of life.

We expected EEG signals to show an increasingly stronger encoding of phonetic feature categories across the first year of life. Previous behavioural data indicate that infant perception becomes more selective towards native than non-native speech contrasts around 9–12 months of age (this should not be interpreted as a hard boundary, as this is likely a gradual phenomenon that changes over large time windows, with differences between easy and more difficult speech contrasts)[42], with perceptual "magnet" effects helping to isolate native from non-native phonetic contrasts already by 6 months[43]. We hypothesised that these phenomena may be underpinned by a progressively more precise and acoustically invariant neural encoding of the phonetic features of their native language. This encoding would be expected to emerge as a neural response to speech that reflects a growing invariance towards phonetic categories, where the limit case would be to have neural responses to phonetic categories that are fully invariant to acoustic changes. This longitudinal investigation enables us to track the emergence of phonetic feature encoding in the first year of life while accounting for the full complexity of nursery rhyme listening environments. Our analysis indicated a progressive increase in encoding over the first year of life, demonstrating that statistically significant invariant encoding emerges from 7 months of age.

## Results

### The neural tracking of phonetic features increases across the first year of life

A multivariate TRF analysis was carried out to assess the low-frequency (0.1–8 Hz) neural encoding of speech across the first year of life. The 8-band acoustic spectrogram of the sound (**S**) and phonetic feature vectors (**F**) were extracted from the stimulus. Fourteen phonetic features were included to mark the categorical occurrence of speech sounds, according to articulatory features describing voicing as well as manner and place of articulation (see "Methods"). Single-participant TRFs were derived for each experimental session to assess the cortical encoding of acoustic and phonetic features by fitting a multivariate lagged regression model for the **S** and **F** features separately (Fig. 1A).

EEG prediction correlations were calculated against ground-truth EEG signals (average EEG across all participants and channels within a group; see "Methods") with leave-one-out cross-validation and averaged across all EEG channels. Prediction correlations were greater than zero for all age groups, models (**S** and **F**), and frequency bands (one-sample two-tailed Wilcoxon rank sum test, FDR-corrected for multiple comparisons, $p < 0.005$). As cortical acoustic-phonetic EEG responses were previously shown to not exceed latencies of 400 ms, this analysis was carried out by considering a slightly larger speech-EEG lag window from −100 to 500 ms to account for possibly longer responses in infants. Negative lags were included for absorbing the mTRF side artefacts, allowing for interpretation of the positive lags. As hypothesised, the EEG tracking of phonetic features (but not of sound acoustics) increased with age (Fig. 1B). The TRF analysis indicated that the EEG prediction correlations for $TRF_F$ increases with age (4mo < 7 mo < 11mo) in the $\Delta$-band (repeated measures ANOVA: $F(2,92) = 4.6$, $p = 0.013$, $\eta_p^2 = 0.091$) and $\Theta$-band (repeated measures ANOVA: $F(2,92) = 5.8$, $p = 0.004$, $\eta_p^2 = 0.112$), while there was no statistically significant effect in the low$\Delta$-band (repeated measures ANOVA: $F(1.95,89.7) = 0.9$, $p = 0.427$, $\eta_p^2 = 0.018$; the assumptions of sphericity was not met; hence, a Greenhouse-Geisser's correction was applied). By contrast, age impacted acoustic tracking in a different way, with 4mo showing the strongest tracking in the low$\Delta$-band (repeated measures ANOVA: $F(1.55,71.2) = 6.9$, $p = 0.002$, $\eta_p^2 = 0.131$; the assumptions of sphericity were not met; hence, a Greenhouse-Geisser's correction was applied) and $\Delta$-band (repeated measures ANOVA:

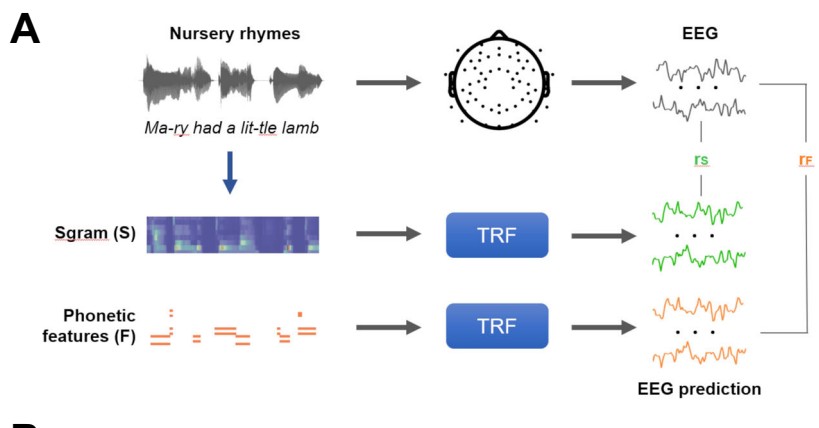

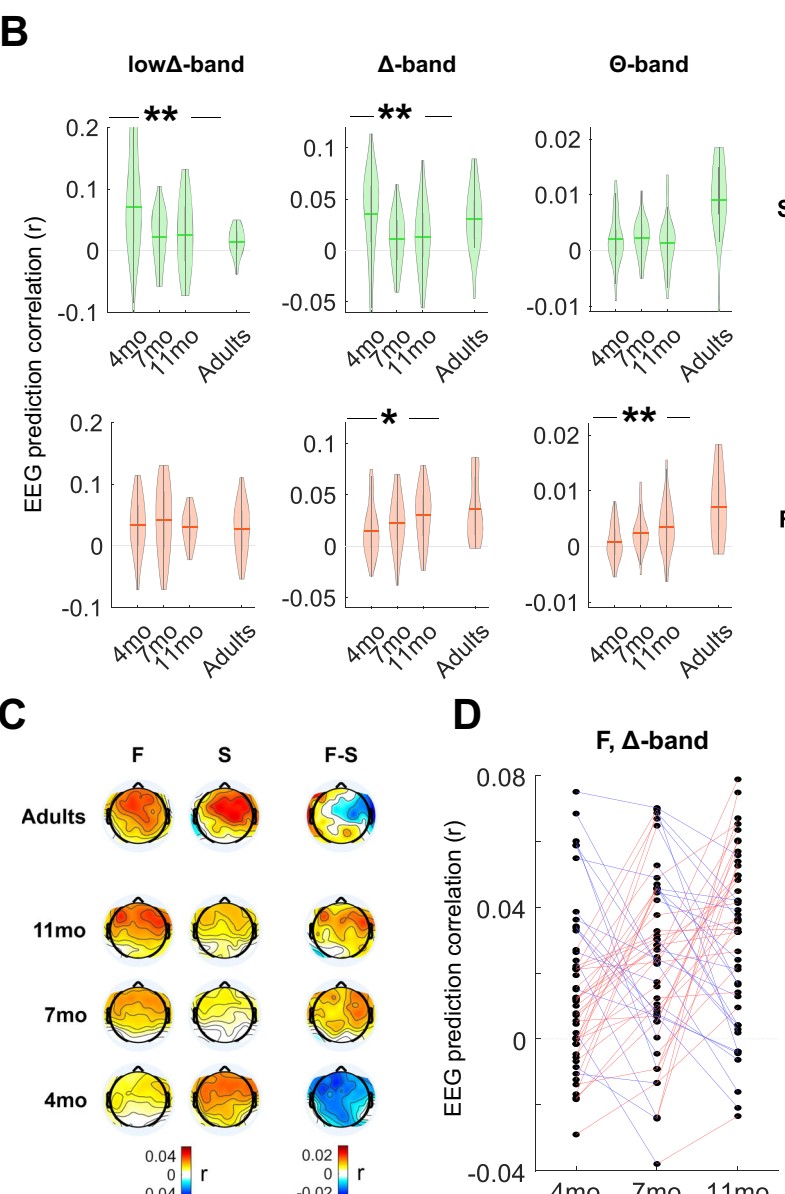

$F(1.74, 80.2) = 6.8$, $p = 0.002$, $\eta_p^2 = 0.129$; the assumptions of sphericity was not met; hence, a Greenhouse-Geisser's correction was applied), while there was no statistically significant effect in the Θ-band (repeated measures ANOVA: $F(2,92) = 0.5$, $p = 0.580$, $\eta_p^2 = 0.011$). Detailed post hoc statistical results are reported in Supplementary Table 1 and Fig. 1C shows individual infant trajectories for TRF$_F$ in the Δ-band (where the encoding of phonetic features shows large r values).

Topographic differences were expected both across participants and by age group due to major anatomical changes during infancy[44]. In the Δ-band, larger EEG prediction correlations were measured in centro-frontal electrodes for all age groups (Fig. 1D), with topographies becoming progressively more similar to those for adults with age (bootstrap with group-size = $N_{adults}$ = 17 and 100 iterations; average correlation with adults: $r = 0.44$, 0.58, 0.60 for 4mo, 7mo, and 11mo

**Fig. 1 | Increasing low-frequency EEG tracking of phonetic features but not sound spectrogram in the first year of life. A** Schematic diagram of the analysis paradigm. Multivariate Temporal Response Function (TRF) models were fit to describe the forward relationship between stimulus features and the low-frequency EEG signal recorded from infants (4, 7, and 11 mo) and adults. TRF models were fit for acoustic spectrogram (**S**; green) and phonetic feature categories (**F**; orange) separately. EEG prediction correlations (Pearson's correlation) were calculated for the $TRF_S$ and $TRF_F$ models with cross-validation. **B** EEG prediction correlations of the $TRF_S$ and $TRF_F$ models (average across all channels) for the low$\Delta$-band (0.1–1 Hz), $\Delta$-band (1–4 Hz), and $\Theta$-band (4–8 Hz; violin plots with mean value

across participants). Stars indicate significant effects of age in infants (one-way repeated measures ANOVA; $*p \leq 0.05$; $**p \leq 0.01$). Statistically significant effects were measured for S in low$\Delta$-band ($p = 0.002$) and $\Delta$-band ($p = 0.002$), but not $\Theta$-band ($p = 0.580$). Statistically significant effects were measured for F in $\Delta$-band ($p = 0.013$) and $\Theta$-band ($p = 0.004$), but not low$\Delta$-band ($p = 0.427$). The violin plots show the result distributions and the mean value. **C** Topographical patterns of the EEG prediction correlations in infants and adults for the F and S models. **D** Individual-participant trajectories of the EEG prediction correlations for the longitudinal infant cohort. Colours indicate increasing vs. decreasing patterns with age (red and blue respectively). The figure was built using MATLAB software.

respectively; since a Shapiro-Wilk test indicated that one of the samples was not normally distributed, a Friedman test was run in place of a repeated measures ANOVA, on infant data with age as the repeated factor: $\chi^2(2,198) = 73.2$, $p < 0.001$, Kendall's W = 0.366).

An additional control analysis was run to determine whether the effect on $TRF_F$ could have been determined by differences in the signal-to-noise ratio (SNR) between groups. The SNR was calculated as the ratio between post- and pre-stim power for the event-related potential calculated on the first word in each trial. There was no statistically significant effect of SNR (since one of the samples was not normally distributed, a Friedman test was run: $\chi^2(2, 92) = 0.2$, $p = 0.917$, Kendall's W = 0.002).

## Emergence of the encoding of phonetic feature categories

The results in the previous section indicate that neural encoding of phonetic feature information in the $\Delta$- and $\Theta$-bands becomes progressively stronger with age in the first year of life. However, it is unclear how much of that effect reflects the emergence of categorical encoding of phonetic features, as $TRF_F$ alone could capture both acoustically variant and invariant neural signals. Accordingly, we carry out further analyses that aim to isolate EEG responses to phonetic feature categories that are acoustically invariant. In line with previous behavioural work[43,45–49] and current developmental theories[1,42], we expected categorical phonetic feature encoding to emerge from 6 months on (i.e., from the 7mo recording session, in the present study), with progressively stronger encoding across the first year of life visible by 11 months of age. To test this hypothesis (see Hp$_2$ in Fig. 2B-left), acoustically invariant phonetic feature encoding was assessed based on a multivariate TRF model that was fit on acoustic-phonetic features simultaneously (acoustic-phonetic TRF). The feature set consisted of **S, F**, and two nuisance regressors: the half-way rectified envelope derivative (**D**) and a signal capturing the overall frame-to-frame visual motion (**V**). A second model was fit after excluding all phonetic features (acoustic-only TRF; Fig. 2A). Neural activity linearly reflecting phonetic feature categories but not sound acoustics was accounted for by subtracting EEG prediction correlations corresponding to acoustic-only TRFs from those corresponding to acoustic-phonetic TRFs. The resulting EEG prediction gain, which was previously referred to as **FS-S**[25,27,33,35,36,50,51], was expected to be most prominent in the $\Delta$-band, in line with previous research on the cortical tracking of speech in adults[25,51] as well as considering that the amplitude modulations of infant-directed speech are maximised in that same frequency-band[52]. Note that the analyses that follow have been carried out only on the EEG frequency bands where $TRF_F$ showed significant effects of age ($\Delta$- and $\Theta$-bands only, while low$\Delta$-band was excluded).

As hypothesised (Hp$_{1-3}$), the EEG prediction gain increased with age in the $\Delta$-band (average across all EEG channels; repeated measures ANOVA: $F(2,92) = 6.0$, $p = 0.003$, $\eta_p^2 = 0.115$; Fig. 2B-right). Statistically significant effects also emerged in the $\Theta$-band (repeated measures ANOVA: $F(2,92) = 5.0$, $p = 0.009$, $\eta_p^2 = 0.098$; Fig. 2B-right). In the $\Delta$-band, consistent with hypothesis 2 (Hp$_2$; Fig. 2B-left), EEG prediction gains greater than zero emerged from 7 months of age (one-sample two-tailed Wilcoxon rank sum test, FDR-corrected; 4mo: $p = 0.554$;

7mo: $p = 0.044$; 11mo: $p = 0.002$; adults: $p = 0.038$). Furthermore, post hoc comparisons indicated a significant increase between 4 and 7mo and 4 and 11mo ($p = 0.041$ and $p = 0.004$ respectively, paired Wilcoxon rank sum test, FDR-corrected), while there was no statistically significant difference between 7 and 11mo ($p = 0.220$). In the $\Theta$-band, EEG prediction gains greater than zero also emerged from 7 months of age (one-sample two-tailed Wilcoxon rank sum test, FDR-corrected; 4mo: $p = 0.7223$; 7mo: $p = 0.006$; 11mo: $p = 0.007$; adults: $p = 0.723$). Furthermore, post hoc comparisons indicated a significant increase between 4 and 7mo and 4 and 11mo ($p = 0.033$ and $p = 0.022$ respectively, paired Wilcoxon rank sum test, FDR-corrected), while there was no statistically significant difference between 7 and 11mo ($p = 0.849$). The regression weights for the acoustic-phonetic TRF model are reported in Fig. 2C (all weights including the nuisance regressors are reported in Supplementary Fig. 1). For ease of visualisation, only the weights for a model fit in the most relevant frequency bands ($\Delta$- and $\Theta$-bands) are reported.

## Discussion

The present investigation offers neurophysiological and longitudinal evidence that the human cortex displays a progressive increase in phonetic encoding during nursery rhyme listening across the first year of life. The results demonstrate significant progress with age (Fig. 1B) as hypothesised a priori, with invariant encoding emerging from 7 months of age (Fig. 2B). Interestingly, phonetic category encoding emerged in both the EEG $\Delta$- and $\Theta$-bands, which is in line with previous work with adults[25]. The largest prediction gains were found in the EEG $\Delta$-band, the frequency range that captures the exaggerated metrical and stress patterns characterising nursery rhymes[52,53]. A fine-grained and longitudinal understanding of the development of phonetic feature encoding by the same infants listening to continuous speech was previously absent from the literature. The behavioural and MMR infant speech processing literature instead relied on targeted experimental contrasts, focused largely on phonetic category formation, but including other manipulations based on syllable stress templates and speech rhythm[17,42–49]. As rhythm and stress patterns aid in identifying word boundaries, and phonetic categories aid in comprehension (e.g., distinguishing 'doggy' from 'daddy'), this prior work has been important, showing that infants are sensitive to differences in speech rhythm from birth[54,55], and are also sensitive to some phonetic information as neonates[56]. Nevertheless, no prior study has used nursery rhymes, nor other continuous speech stimuli with a rich phonological inventory, as a basis for studying the neurophysiology of phonetic encoding in infants. Consequently, our findings have several implications for developmental research.

The current study demonstrates that we can now use ecologically valid neural data from typical pre-verbal scenarios such as nursery rhyme listening to study different aspects of speech processing by infants. Our continuous speech analyses indicated that phonetic categories are encoded in an acoustically invariant manner from 7 months, thereby also documenting at what stage of development encoding becomes robust. This longitudinal question previously remained open largely because of methodological constraints. The present study offers neurophysiological evidence regarding the

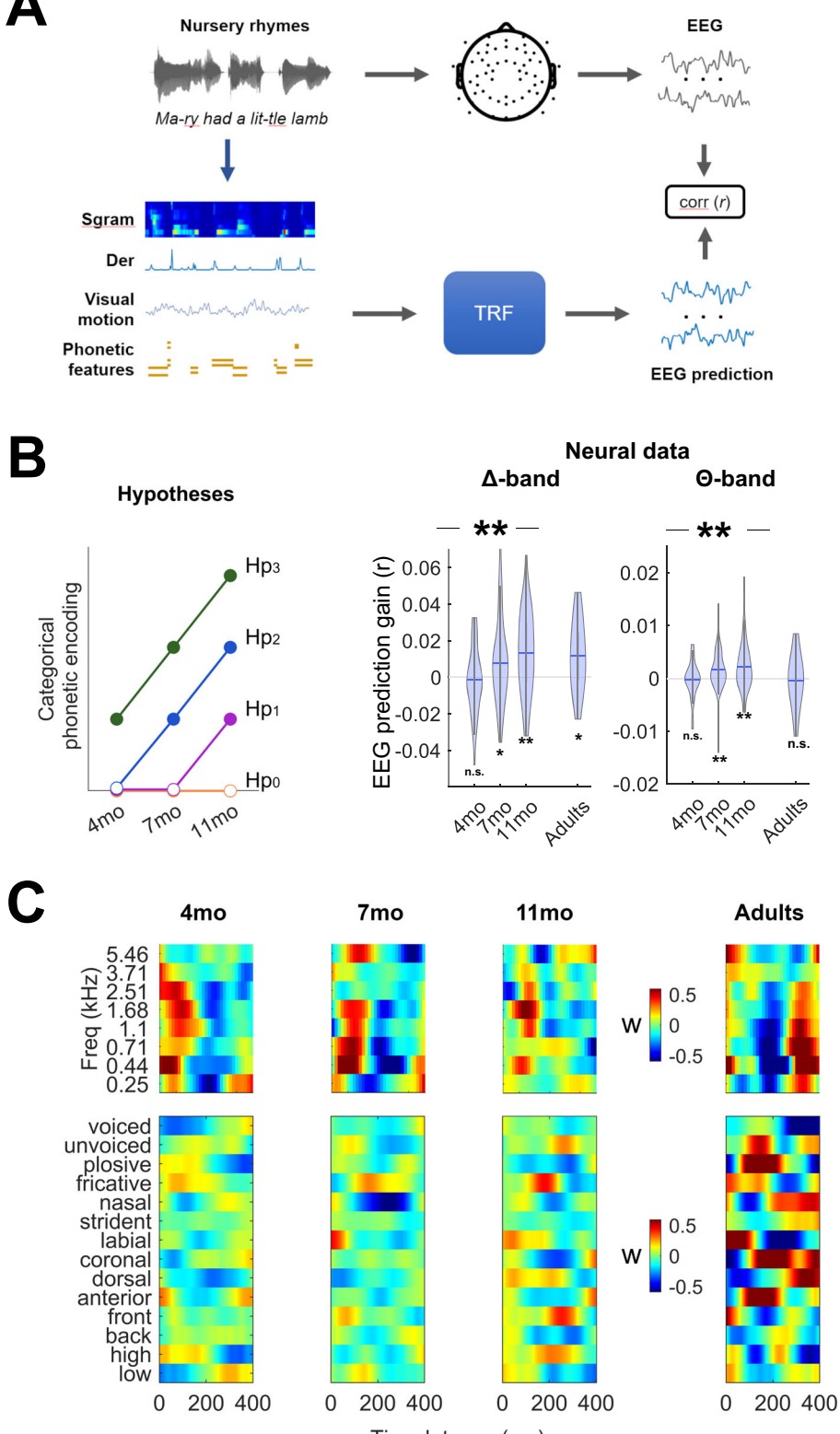

precise progression of invariant phonetic category learning. There is a consensus in the literature that discriminating phonetic categories is a key processing step regarding speech comprehension by adults[57], although see Feldman et al. [58], for recent caveats regarding infants. While adult studies used direct invasive recordings to measure the cortical encoding of phonetic categories[37,59], here we demonstrated that recent methodological developments (i.e., the TRF framework[25,38,39]) can be used to circumvent some of the major challenges encountered by previous infant studies, thereby providing developmental insights that complement behavioural research.

The assessment of phonetic encoding as operationalised here has three key properties. First, we studied the cortical encoding of phonetic categories in infants with continuous neurophysiological measurements (TRFs) based on EEG and as part of an unprecedented targeted longitudinal investigation. Second, the use of the forward TRF framework allowed us to assess phonetic category encoding, rather

**Fig. 2 | Low-frequency (Δ-band) cortical encoding of categorical phonetic features in the first year of life. A** Schematic diagram of the analysis paradigm. Forward TRF models were fit to describe the relationship between speech features (including nuisance regressors) and low-frequency EEG signals. Speech features included the acoustic spectrogram (**S**), half-way rectified envelope derivative (**D**), visual motion (**V**), and phonetic features (**F**). **B** (Left) Hypotheses: The cortical encoding of phonetic feature categories was expected to progressively increase across the first year of life. Hp1-3: phonetic encoding emerging from 11, 7, and 4 months of age respectively. (Right) Phonetic feature encoding measured as the EEG prediction correlation gain when including phonetic features in the TRF (violin plots with mean value across participants). Only the frequency bands showing significant effects of age for TRF_F were studied. Stars indicate significant effects of age in infants (one-way repeated measures ANOVA; *$p \leq 0.05$; **$p \leq 0.01$; ***$p \leq 0.001$). Statistically significant effects were measured in the Δ-band ($p = 0.003$) and in the Θ-bands ($p = 0.009$). **C** TRF weights corresponding to phonetic features for the TRF in Δ- and Θ-bands (1–8 Hz). The figure was built using MATLAB software.

than relying on the typical sound discrimination metrics used in prior behavioural[43,45–49] and neurophysiology studies (e.g., MMR)[5,60–64]. Third, the TRF framework allowed us to study neural encoding in an ecologically natural paradigm of singing nursery rhymes to infants, instead of focusing on selected phonetic, syllabic or word contrasts or using synthesised speech stimuli, as in the past literature. This is a step forward, as the discriminatory skills that infants were shown to exhibit with isolated syllables may not enable the detection of phonetic categories in continuous speech.

The present study indicates that phonetic category encoding during natural speech listening progressively increased during the first year of life. This provides the literature with detailed insights into the development of speech processing in neurotypical infants. The enhanced phonetic encoding with age observed here could not simply be due to stronger acoustic encoding, as acoustic encoding showed a different pattern in the Δ-band (Fig. 1B). Interestingly, 4mo infants showed the strongest acoustic encoding and the weakest phonetic encoding, with statistically significant phonetic encoding emerging only from 7 months of age. We did not find credible evidence for acoustically invariant phonetic encoding at 4mo, which is in contrast with the significant evidence for phonetic encoding in neural studies with 3-month-olds based on single syllables like "bif" and "bof"[64], which have claimed that very young infants are equipped with the fundamental combinatorial code for speech analysis. While prior demonstrations of behavioural and neural discrimination between syllables in infants younger than 4mo may reflect categorical phonetic encoding, it is also possible that successful discrimination in these simplified tasks may have an acoustic basis. In other words, the ability to distinguish two sounds does not necessarily mean that those sounds are encoded as separate categories during natural speech listening. Another developmental possibility is that categorical encoding does occur during simplified tasks, but this does not generalise to more complex tasks like nursery rhyme listening, which requires greater cognitive resources. By probing the neural encoding of phonetic categories directly, our study provides the field with a valuable platform for addressing these unanswered questions and for better understanding the cognitive processes underlying speech perception in infants.

One challenge with longitudinal neurophysiology studies in infants is the substantial anatomical change that occurs with age, meaning that while macroscopic patterns are likely to remain consistent (e.g., temporal vs. occipital), there cannot be a channel-by-channel correspondence between age groups, even when considering the same participants. For this reason, the majority of this investigation focused on measures combining multiple EEG channels simultaneously (e.g., Fig. 1B was an average of all EEG channels). These considerations strengthen the results, as averaging all the EEG channels would not perform as well as selecting the best electrode for each participant. As such, our statistical analyses were conservative in that they were penalised by the inclusion of all EEG channels, even the ones that were not responsive to speech.

Our phonetic encoding results showed topographical patterns for adults that are broadly consistent with the prior adult EEG literature on natural speech listening TRFs (Fig. 1D)[25,33]. Indeed, some differences with prior adult work could be expected, as this TRF investigation of phonetic processing relied on a nursery rhyme listening task rather than natural speech. Nursery rhymes are indeed a form of natural speech which is more suited to infants, and would naturally be delivered audio-visually. The rhythmic cues and exaggerated stress patterns characterising nursery rhymes have been demonstrated to be important elements supporting speech perception and language learning[52,65], accordingly, they were ideal stimuli for the Cambridge UK BabyRhythm study. In prior TRF work, we have demonstrated similar envelope entrainment to these nursery rhymes by adults and infants[24]. Nevertheless, it is important to note that the regular rhythms and melodic properties of nursery rhymes make them different from the typical speech TRF stimuli used with adults, such as audio-books and podcasts, and that encoding of phonetic categories for auditory-alone continuous speech to infants remains to be investigated. Another difference from previous work on phonological TRFs is the use of audio-visual stimuli. Note that the visual stimulus could not explain the finding (as removing the visual motion regressor did not change the results).

The results of this study add to the growing literature on cortical speech tracking[19,25,27–30,36,53]. While the literature typically focuses on the cortical tracking of the speech envelope[14,22,66–69] (including previous analyses of this dataset[19,24]), the present investigation enriches our understanding of phonetic feature TRFs. Prior TRF studies of phonetic encoding in adults and children have revealed that phonetic processing is affected by speech clarity[51], selective attention[33], and proficiency in a second language[36], and shows correlations with psychometric measures of phonemic awareness[35]. The present study demonstrates that emergent phonetic TRFs can also be measured in pre-verbal infants, providing a window into the neural encoding of nursery rhymes in infants. Whilst recent developments have started to use neural tracking to predict language development in infants[70,71], further research will also determine whether a robust relationship exists between speech TRFs and other related aspects of cognition (e.g., selective attention, prediction) in infants, and when such related aspects come online. Further research with infants at family risk for disorders of language learning may also reveal when and how developmental trajectories are impacted by developmental disorders that are carried genetically, such as developmental dyslexia and developmental language disorder. Such work could be very valuable regarding early detection and improved mechanistic understanding of these disorders.

In summary, the data provide evidence of the emergence of phonetic categories that contribute to the current debate regarding their role in the development of speech processing[58]. Our demonstration that phonetic encoding can be assessed with nursery rhyme stimuli in ecologically valid conditions opens the door to cross-language work using TRFs that investigates the interaction between characteristics of natural language such as phonological complexity and the development of phonetic encoding. It also provides opportunities for novel mechanistic investigations of the development of bilingual and multilingual lexicons during language acquisition.

## Methods
### Participants and experimental procedure
The present study carried out a re-analysis of an EEG dataset involving a speech listening task in a longitudinal cohort of fifty infants (first part of a larger cohort of 122 participants[19]). The first 50 participants of that

longitudinal cohort who were able to provide data for all three sessions were included. Participants were infants born full term (37–42 gestational weeks) and had no diagnosed developmental disorder, recruited from a medium-sized city in the United Kingdom and surrounding areas via multiple means (e.g., flyers in hospitals, schools, and antenatal classes, research presentations at maternity classes, online advertising). The study, including experiments on adults and infants, was approved by the Psychology Research Ethics Committee of the University of Cambridge. Parents gave written informed consent after a detailed explanation of the study and families were repeatedly reminded that they could withdraw from the study at any point during the repeated appointment. The experiment involved three EEG recording sessions when the infants (24 male and 26 female) were 4 months old (4mo; 115.6 ± 5.3 days), 7 months old (7mo; 212.5 ± 7.2 days) and 11 months old (11mo; 333.0 ± 5.5 days) [mean ± standard deviation (SD)]. A bilingualism questionnaire (collected from 45 out of the 50 infants) ascertained that 38 of the infants were exposed to a monolingual environment and 12 were exposed multilingual environment, of these 93.5% (43 infants) reported English as the primary language exposed to the infant. Note that this was a longitudinal investigation, meaning that the same 50 infants were tested at 4, 7, and 11 months of age. In addition to the 150 EEG sessions from the infant dataset, this study also analysed EEG data from twenty-two monolingual, English-speaking adult participants performing the same listening task (11 male, aged 18–30, mean age: 21), who gave written informed consent. Data from five adult participants were excluded due to inconsistencies with the synchronisation triggers, leaving 17 participants' data for the analysis. We did not expect an effect of sex on phonetic-feature TRFs in the first year of life.

Infant participants were seated in a highchair (one metre in front of their primary caregiver) in a sound-proof acoustic chamber, while adult participants were seated in a normal chair. All participants were seated 650 mm away from the presentation screen. EEG data were recorded at a sampling rate of 1 kHz using a GES 300 amplifier using a Geodesic Sensor Net (Electrical Geodesics Inc., Eugene, OR, United States). 64 and 128 channels were used for infants and adults respectively. Sounds were presented at 60 dB from speakers placed on either side of the screen (Q Acoustics 2020i driven by a Cambridge Audio Topaz AM5 Stereo amplifier). Participants were presented with eighteen nursery rhyme videos played sequentially, each repeated 3 times (54 videos with a presentation time of 20' 33" in total). Adult participants were asked to attend to the audio-visual stimulus while minimising their motor movements. All adult participants completed the full experiment. Infants listened to at least two repetitions of each nursery rhyme (minimum of 36 nursery rhymes lasting 13' 42"). The experiment included other elements that were not relevant to the present study (e.g., resting state EEG; please refer to the previous papers on this dataset for further information[19,24]). To measure the time each infant looked at the screen, eye tracking data were collected using a Tobii TX300 eye tracking camera (sampling rate 300 Hz) located and fixed at the base of the presentation screen (23" TFT monitor).

## Stimuli

A selection of eighteen typical English language nursery rhymes was chosen as the stimuli. Audio-visual stimuli of a singing person (upper-body only) were recorded using a Canon XA20 video camera at 1080p, 50fps and with audio at 4800 Hz. A native female speaker of British English used infant-directed speech to melodically sing (for example "Mary Quite Contrary") or rhythmically chant (for nursery rhymes like "There was an old woman who lived in a shoe") the nursery rhymes whilst listening to a 120 bpm metronome through an intra-auricular headphone (e.g., allowing for 1 Hz and 2 Hz beat rates; see Supplementary Figs. S2 and S4 from Attaheri et al.[19]). The metronome's beat was not present on the stimulus audios and videos, but it ensured that a consistent rhythmic production was maintained throughout the 18 nursery

rhymes. To ensure natural vocalisations, the nursery rhyme videos were recorded sung or rhythmically chanted, live to an alert infant.

## Data preprocessing

Analyses were conducted with MATLAB 2021a by using custom scripts developed starting from publicly available scripts shared by the CNSP initiative (Cognition and Natural Sensory Processing; https://cnspworkshop.net; see section Data and Code Availability for further details)[72]. In order to carry out the same preprocessing and analysis pipeline on infants and adult EEG data, the adult 128-channel EEG data was transformed into a 64-channel dataset via spline interpolation, with the relative channel locations corresponding to those of the infant participants. All subsequent analyses on infants and adults were identical.

The four facial electrodes (channels 61–64) were excluded from all analyses, as they are not part of the specific infant-sized EGI Geodesic sensor net. The EEG data from the remaining 60 channels were low-pass filtered at 8 Hz by means of zero-phase shift Butterworth filters with order 2 (by using the filtering functions in the CNSP resources) to include the Δ- and Θ-bands, which were shown to strongly encode cortical auditory responses to natural speech[19,25]. EEG data were also high-pass filtered at 0.1 Hz to reduce noise and downsampled to 50 Hz. Next, Artifact Subspace Reconstruction (ASR; clean_asr function from EEGLAB[73]) was used to clean noise artefacts from the EEG signals. Channels with excessive noise (which could not be corrected with ASR) were identified via probability and kurtosis and were interpolated via spherical interpolation if they were three standard deviations away from the mean. EEG signals were then re-referenced to the average of the two mastoid channels, which were then removed from the data, producing a preprocessed EEG dataset with 58 channels. Data from repeated trials was then averaged. Three infant participants were removed because of excessive noise in at least one of their three recording sessions. Average EEG signals were obtained by averaging data from all EEG channels and participants, within each group, leading to a single EEG trace per age group, which we refer to as ground-truth EEG. Note that this was possible due to the mastoid referencing (i.e., averaging across electrodes would not work when using global average referencing). Ground-truth EEG traces were obtained for three frequency bands of interest: the lowΔ- (0.1–1 Hz), Δ- (1–4 Hz) and Θ- (4–8 Hz) bands and used in the TRF evaluation.

## Sung speech representations

The present study involved the measurement of the coupling between EEG data and various properties of the sung speech stimuli. These properties were extracted from the stimulus data based on methodologies developed in previous research. First, we defined a set of descriptors summarising low-level acoustic properties of the speech stimuli. Acoustic features consisted of an 8-band acoustic spectrogram (S) and a half-way rectified broadband envelope derivative (D)[36,53]. S was obtained by filtering the sound waveform into eight frequency bands between 250 and 8 kHz that were logarithmically spaced according to the Greenwood equation[74]. The broadband envelope was calculated as the sum across the eight frequency bands of S. The D signal was then derived by calculating the derivative of the broadband envelope, and by half-way rectifying the resulting signal. Second, fourteen phonetic features were selected to mark the categorical occurrence of speech sounds, according to articulatory features describing voicing, manner, and place of articulation[75,76]: voiced consonant, unvoiced consonant, plosive, fricative, nasal, strident, labial, coronal, dorsal, anterior, front, back, high, low, leading to a 14-dimensional phonetic feature categories matrix (F). The precise timing of the phonetic units was identified in three steps. First, syllable and phoneme sequences were obtained from the transcripts of the nursery rhymes. Second, an initial alignment was derived by identifying the syllabic rate and syllable onsets for each piece and then assigning the

phonemes in a syllable starting from the corresponding onset time. This automatic alignment was stored according to the TextGrid format[77]. Third, the phoneme alignments were manually adjusted using Praat software[77]. Phonetic feature vectors were produced in MATLAB software to categorically mark the occurrence of phonetic units from start to finish with unit rectangular pulses[25]. Next, phonetic units were mapped to the corresponding 14-dimensional phonetic feature vectors, leading to a stimulus matric with a reduced dimensionality (14 features instead of 27 phonemes). Finally, a nuisance regressor was also included to capture EEG variance related to visual motion (**V**), which was derived as the frame-to-frame luminance change, averaged across all pixels.

### Multivariate temporal response function (mTRF)

A single input event at time $t_O$ affects the neural signals for a certain time window $[t_1, t_1+t_{win}]$, with $t_1 \geq t_O$ and $t_{win} > 0$. Temporal response functions (TRFs) describe this relationship at the level of the individual participant and EEG channel. In this study, TRFs were estimated by means of a multivariate lagged regression, which determines the optimal linear transformation from stimulus features to EEG (forward model)[11,78]. A multivariate TRF model (mTRF) was fit for each participant by considering all features simultaneously (**S**, **D**, F, and **V**; Fig. 1A) with the mTRF-Toolbox[38,39]. While previous work used a time-lag window of 0–400 ms, which was considered sufficient to largely capture the acoustic-phonetic/EEG relationship with a single-speaker listening task in adults[25], the relevant latencies in infants were unknown before this study. To account for possible slower or delayed response in infants, a larger time-latency window of −100 to 500 ms was used in the analysis for all groups. The reliability of the TRF models was assessed using a leave-one-out cross-validation procedure (across trials i.e., nursery rhymes), which quantified the EEG prediction correlation (Pearson's $r$) on unseen data while controlling for overfitting. The TRF model calculation included a Tikhonov regularisation, which involves the tuning of a regularisation parameter ($\lambda$) that was conducted by means of an exhaustive search of a logarithmic parameter space from 0.01 to $10^6$ on the training fold of each cross-validation iteration[38,39]. Note that the correlation values are typically calculated between EEG signals and their predictions by considering single-participant EEG signals, which have a high level of noise. As such, EEG prediction correlations are variable between participants largely due to the variable SNR of the EEG signal across participants (as every prediction is correlated with a different EEG signal). One solution to this issue is, instead, to correlate the EEG predictions for each participant with the same *ground-truth* EEG trace calculated as described in the "Data preprocessing" section.

### Statistical analysis

All statistical analyses directly comparing the groups were performed using repeated measures ANOVA, with $F$-values reported as $F(\text{df}_{time}, \text{df}_{error})$ when the assumptions of normality and sphericity were met. Those assumptions were tested with Shapiro-Wilk's test and Mauspher's test respectively. Assumptions of normality for statistical tests were met unless otherwise stated. When the assumption of sphericity was not met, a Greenhouse-Geisser's correction was applied. When the assumption of normality was not met, a Friedman test was applied. Two-tailed one-sample Wilcoxon signed-rank tests were used for post hoc tests. Correction for multiple comparisons was applied where necessary via the false discovery rate (FDR) approach. When post hoc comparisons were carried out for multiple models and frequency bands, the correction took into account all the comparisons simultaneously. The FDR-adjusted $p$-value was reported. Descriptive statistics for the neurophysiology results are reported as a combination of mean and standard error (SE).

### Reporting summary

Further information on research design is available in the Nature Portfolio Reporting Summary linked to this article.

### Data availability

The present study carried out a re-analysis of an existing EEG dataset recorded from infants and adults[19]. All EEG data was converted to the CND data structure (Continuous-event Neural Data https://cnspworkshop.net, version 2023.0)[72], allowing to carry out the analyses with the CNSP analysis scripts, which provided a platform for bringing together all the necessary libraries. The final data included in the manuscript figures and statistics have been deposited in the OSF repository https://osf.io/mdnwg[79]. The raw data are not downloadable as they must be considered in conjunction with the data collection videos. As the participants are infants, these video data are confidential data that we do not have ethical permission to make available. Study data were collected and managed using REDCap (Research Electronic Data Capture) electronic data capture tools hosted at Cambridge university[80,81]. The raw EEG data, without the video data, are available upon request (please contact the corresponding author).

### Code availability

Analyses were conducted by using the publicly available analysis scripts shared by the CNSP initiative (Cognition and Natural Sensory Processing)[72] during the CNSP workshop 2021 (https://cnspworkshop.net). The scripts were modified to fit this particular study. The specific analysis scripts have been deposited in the OSF repository https://osf.io/mdnwg[79]. The exact figures in this manuscript can be replicated with the code and data provided. Note that the code utilises external publicly available libraries: the mTRF-Toolbox (https://github.com/mickcrosse/mTRF-Toolbox, version 2.0)[39], EEGLAB (version 2021.0)[73]; and the NoiseTools library (http://audition.ens.fr/adc/NoiseTools)[82].

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

## Acknowledgements

We thank Dimitris Panayiotou, Alessia Philips, Natasha Mead, Helen Ola-wole-Scott, Panagiotis Boutris, Samuel Gibbon, Isabel Williams, Sheila Flanagan, and Christina Grey who helped collect the data as well as all the families of the infant participants. We thank Dr. Susan Richards for her assistance on the phoneme transcription. We thank the CogHear work-shop organisers (Mounya Elhilali, Malcolm Slaney, and Shihab Shamma) and participants for their useful feedback on the early results of this study. This project received funding from the European Research Council (ERC) under the European Union's Horizon 2020 research and innovation pro-gramme (Grant Agreement No. 694786 to U.G.) (A.A., A.N.C., S.R., P.B., U.G.). This research was conducted with the financial support of Science Foundation Ireland under Grant Agreement No. 13/RC/2106_P2 at the ADAPT SFI Research Centre at Trinity College Dublin (G.D.L., G.C.). ADAPT, the SFI Research Centre for AI-Driven Digital Content Technology, is fun-ded by Science Foundation Ireland through the SFI Research Centres Programme. This work was also supported by the Science Foundation Ireland Career Development Award 15/CDA/3316 (G.D.L., R.R.). G.C. was supported by an Advanced European Research Council grant (NEUME, 787836) and by the FrontCog grant ANR-17-EURE-0017.

## Author contributions

U.G., A.A. and G.D.L. conceived the study. A.A. and P.B. programmed the task. A.A., A.N.C., S.R., P.B. and the BabyRhythm Team collected the data. A.A. and G.D.L. preprocessed the data. G.D.L. and G.C. analysed the data. G.D.L., A.A. and U.G. wrote the first draft of the manuscript. G.C., R.B.R., A.N.C., S.R. and P.B. edited the manuscript.

## Competing interests

The authors declare no competing interests.
