## [Peer Review File · Nature Communications]

Emergence of the cortical encoding of phonetic features in the first year of lifeReviewers' comments:

Reviewer #1 (Remarks to the Author):

Understanding how infants process the features unique to speech and identifying in what ways and when infants' brain responses to speech resemble those of adults is one of the deep challenges in developmental cognitive neuroscience. The Liberto et al. manuscript reports a substantial advance in what we know about infant speech perception with this set of results. They use a novel method that combines cortical tracking of speech in infancy to investigate the emergence of phonetic feature encoding, and they use natural speech (nursery rhymes), which has never been done before. Whole-head EEG was used to examine cortical tracking, and the longitudinal design tested infants at 4, 7, and 11 months, as well as adults. The statistical analysis of the multivariate data produced by temporal response function (TRF) analysis is sophisticated. The results reveal a beautiful progression from 4-11 months, with ever-increasing precision of the neural encoding of phonetic features in speech. Importantly, non-speech control stimuli did not produce this change over time in the same babies. Phonetic feature categories are reflected in the neural data at both 7 and 11 months of age, but not at 4 months of age, and also reveal that at 11 months, infants have not yet reached adult neural processing (an important point). This is a benchmark study that will be a springboard for further work both on typically developing and at risk infant populations. It reaches, in my view, the level of innovation required for Nature Communications.

Reviewer #2 (Remarks to the Author):

The authors apply an analytical EEG approach by the first author to a longitudinal EEG data set of 4 mo, 7 mo and 11 mo old infants.

The authors claim that these data demonstrate for the first time how phonetic encoding emerges cortically, and that it does emerge or has emerged at 7 months of age. The study is rich in detail, but offers, at least in its current form, a string of somewhat frustrating analytical choices or shortcomings that leave open how justified some of the more bold conclusions actually are. A major concern in my view is that phonetic encoding emerging in the first year of life has been amply demonstrated in arguably more compelling ways before.

I have organised my further comments below by perceived severity:

-- l. 69: Claims of primacy are usually problematic: I would simply not agree with this study being the "first to address the research questions directly"? Yes, the linear model obviously offers the advantage of tackling continuous speech, as the first author and many, many authors have demonstrated before, also in infants. This is a technical question, however. As for the two other question, "how are speech sounds encoded" and how speech sound representation develops over the first year of life, psycholinguist and phonological work has offered tremendous insight over the last decades. As such, I see primarily an advance in technical fidelity here, by applying the Di Liberto 2015 analytical framework (in short, 'what do phonetic features add to an acoustic model of cortical speech encoding?') to infant data.

-- The arguably most important results (and the results most likely to motivate a large-scale, high-impact publication) are of borderline conventional significance: In lines 169f. The authors argue for phonetic-features information improving model fits/predictive accuracy in 11-mo and 7-mo olds (but not in 4-mo olds) with FDR-corrected p values all in the [0.023;0.05] range.

-- Furthermore, the n.s. results for 4-mo olds are not given, but to my reading a formal comparison that would show that these results in 7 mo and 11 mo are indeed superior to those in 4 mo olds is missing. This would constitute an inference fallacy but might well be (and would need to be) rectified in revision.

-- In line 185f., in the very laudable canonical correlation analysis the authors turn to, it is notable that only the restricted time window yields a (borderline) significant effect, while the time window having yielded all previous effects more somewhat more substantially is far from significant.

Here again, a look at the TRF would have been helpful to understand the underlying response rather than the overly condensed predictive-accuracy measure only.

Also here, the so pivotal age-group comparison remains very cursory and qualitative: We are assured of significant effects at "large clusters of electrodes" from "7 months on", but again a more formal and stringent comparison of topographies by age groups is missing. It can thus not be known how meaningful the change in significant channels is.

This particularly relates/leads to my other main concern outlined, on the lurking/unknown role of signal/noise ratio in this analysis and conclusion.

-- p. 11/Discussion: As stated in other terms in other instances of my review, the conclusion offered ("first evidence that the human cortex encoded phonetic categories during the first year of life") is overly strong: There can hardly be any doubt that the work by Werker, Kuhl, Näätänen and others (major uncited paper eg Stager & Werker, Nature 1997) has demonstrated a striking change in phonetic category perception between approx. 4 and approx. 11 months of age. I applaud the approach to use the di Liberto "phonetic-features plus acoustics" modelling approach to infant data, but the new insight appears to be mainly of technical nature. Moreover, the data themselves leave worries over how robust the claimed "gradual" change in predictive accuracy, TRF change, and topography really is.

The authors go at great length in the discussion justifying the novelty of their findings. They argue in particular that the present study is first to elucidate the precise time point. For reasons developed in my more technical comments, I am not sure that the data manages to demonstrate such a "pivot point" between 4 and 7 months. I do think that a previous uncited study also from UCL (Iverson lab, Scientific Reports, 2017 <https://www.nature.com/articles/s41598-019-55085-y>) has done this, using what the present authors call "sound discrimination metrics", somewhat more compellingly (as they tackle the emergence of the 'vowel space' quite directly and with compelling age group x vowel-pair interactions). I am just giving this example in somewhat more detail to illustrate how the current paper overemphasises novelty in places where the analyses potentially don't live up to these claims.

-- Lastly, I agree with the authors that a difference in SNR is unlikely to explain all effects, as the acoustic-envelope impact seems not too much affected by age group. However, there is a limit to this argument: As in most models using the di Liberto approach (incremental models of acoustic plus phonetic features), the phonetic features can only add explaining a certain, relatively minor remaining part of the variance. If there were a systematically lowered SNR in the younger EEG data, I would expect the relatively most impactful regressors (acoustic) to work fine nevertheless but more subtle effects to become invisible first (such as phonetics). In short, a more explicit demonstration that SNR is not a worry at all when arriving at this paper's conclusions would be in order.

MINOR:

-- Related to major points above, the hypothesis (spelled out in line 98f) seems to be that "neural encoding of phonetic features during natural speech listening is already developing during the first-year of life". This strikes me as an odd place to start, given that in sentence 1 of the abstract we already learn of this very fact. See other point on claimed novelty.

-- l. 130 and many other instances: p values are not a measure of effect size. Please also report meaningful effect sizes, here, average r_{predict} . If in-text the information gets too dense, I consider the r more informative than the p value.

-- Please allow the reader to judge the TRF model/transfer function to be seen. Previous research tells

us of substantial differences in infant ERPs, and accordingly also in infant TRFs. TRFs (including informative baseline/negative time lags) should be included for all age groups, as any model prediction as a whole obviously depends on the set of weights (ie, the TRF) creating such a prediction. (I later saw that Figure 2 contains some featurally resolved TRFs in image plots; I here referred to more conventional, ERP-like temporal response functions for the acoustics. What Figure 2 resolves, are “phonetic-temporal receptive functions” of sorts, no?)

-- The lack of individual data in Figure 1B and in Figures 2–3 (see note on Figure numbering) is not state of the art. Please allow your readers an impression of the actual variability in these data rather than only the SEM. (Note that I infer the error metric here; the figure caption lacks details). This is especially warranted as these are longitudinal/dependent data! A great deal of information is absent at least from the figures by not emphasising/displaying the longitudinal trajectories of individual infants (plus the aggregate/average, of course).

-- A misnumbering of figures has happened, the label Figure 2 is used twice.

-- Open science/open data: I was missing a statement on how the authors plan to make these data and models publically available?

Reviewer #3 (Remarks to the Author):

SUMMARY

The authors investigate phonetic feature encoding in infants (ages 4, 7 and 11 months) using EEG, and compare this to adult data. They test the hypothesis that phonetic encoding emerges with development, by comparing the unique explained variance of a phonetic feature model above a spectrogram/envelope acoustic model. They find that, indeed, a phonetic model explains additional variance as a function of age, and that the organisation of feature responses (relative dissimilarity) also becomes more adult-like. The authors conclude that phonetic feature encoding emerges during the first year of life.

Overall, I commend the authors on a true data collection tour de force, in order to answer an interesting research question. However, there are a number of experimental decisions that were omitted or not fully motivated in the main manuscript, and some overstepping of claims, which I would need to see revised before recommending this manuscript for publication.

MAJOR

[1] Misleading description of stimulus and task in main text

In the main text of the manuscript, the authors emphasise that they are investigating responses to “continuous natural speech”; however, it was not until I read the methods that I realised they actually used (i) audio-visual materials with (ii) sung speech. This was very hidden away, almost misleadingly so. Of course the accompaniment of visual speech cues and the use of an extremely prosodic stimulus will have a big impact on speech encoding, but this is not properly addressed in the manuscript, other than a small comment in the discussion.

To address this, I believe that the authors need to:

- make it clear in the main text (introduction) that they are using audio-visual sung speech
- provide an explicit analysis to show that this is representative of speech processing without visual accompaniment
- show that this is representative of true natural, non-sung, speech
- provide a demonstration that the gain in their model performance is not due to the more robust integration of audio and visual information with age

[2] Signal to noise ratio as an alternative explanation of results

I feel that an obvious counter explanation of the present results is that the signal to noise ratio of the datasets will differ at different ages. I imagine that the data quality improves as a function of age, with knock-on consequences for the modelling results, but I did not see this issue addressed in the manuscript. Furthermore, as infants develop, their sensory systems improve with age, also. So from the signal perspective, it is likely that the older infants' auditory cortices receive a more robust representation of the stimulus input than the younger infants.

This issue is exacerbated by the lack of a baseline stimulus or baseline analysis. I understand that the main analysis is a subtraction of model fits, of the phonetic+acoustic model above the acoustic model, but it would be helpful to see the model performance curves of each model going into that subtraction, and well as basic visualisation of the resulting TRF kernels.

Furthermore, it is possible that a less noisy neural signal naturally favours the more sparse feature model over the complex spectrogram model. It would be helpful to test this empirically, for example by simulating results under different noise conditions.

[3] Research questions

[a] In a number of places in the introduction, and again in the statement of 3 research questions (pg 3), the authors allude to investigating how speech sounds are "perceived" by infants. For example, research question 1 is "How do infants perceive and encode..." — the task is passive listening, and so I ask the authors to be careful not to describe questions regarding how neural encoding relates to the eventual perception.

[b] Furthermore, if we remove the "perception" part of the first research question, the first two questions become extremely redundant: "how infants encode phonological units" and "how are these speech sounds encoded" seem the same to me.

[c] In the final paragraph of the introduction, the authors restate their question as "Test the hypothesis that the neural encoding of phonetic features during natural speech listening is already developing during the first year of life" — this is a vague hypothesis, and the way it is stated, I am not sure that the outcome could be any other way.

Overall, I ask the authors to be more pinpointed in their research questions, and the true hypothesis space that they are investigating.

[4] Analysis decisions lacking motivation

[a] Why was 1-15 Hz chosen for the analysis?

[b] What is the motivation for separating stressed and unstressed syllables? Wouldn't a true phonetic model be invariant to that distinction?

[5] Window selection

The time-window of 0-400 ms is extremely long considering that the phoneme duration is only about 100 ms, and the latency of phonetic encoding is typically around 100 ms. What long-latency responses are the authors expecting to capture with their encoding models? As the window increases in duration, the authors will be capturing more and more statistical structure of the neighbouring sound sequences, which seems problematic for the current research questions if the aim is to test instantaneous phonetic encoding.

Further to this point, the authors also provide an analysis using 100-500 ms, but there is no explicit comparison of whether the additional 100 ms provides explanatory power. It seems to me that the proper way of testing whether responses are longer lived in infants would be to see if a 0-500 ms model delay provides unique variance above the 0-400 ms model; if not, then do not use a longer window; if it does, then only use the longer window.

MINOR

Speech TRFs do not necessarily reflect neural entrainment (pg4, line83) — they can also just reflect instantaneous evoked responses. Please remove

Over-use of bar plots — should show the underlying distribution of the data

Adults were recorded with a higher density cap, and I imagine that their spline procedure reduces density while retaining signal. This brings the concern that the adult data is inherently of a higher SNR, simply because of the difference in hardware.

Reviewer #4 (Remarks to the Author):

This paper uses multichannel EEG coupled with multivariate Temporal Response function (TRF) modeling to examine brain-derived responses to phonemes in speech in 4-, 7-, & 11-month-olds, and in adults.

While I generally like the research direction and the experimental work, the motivation for the study is somewhat bombastic given the state of neuroscience today. Besides, there are a few, somewhat important methodological considerations that call the results into question. I recommend a revision, and detail my concerns below.

1. The authors state early on (pg. 3 [always referring to the PDF pages of the merged ms]):
“However, behavioural methods can only serve as an indirect index of the emergence of linguistic skills, and cannot reveal when the phonetic encoding in the human cortex becomes invariant across different instantiations.”

And, in the next paragraph,

“This study is the first to address these research questions directly.”

And reiterated in the Discussion section (pg.12):

“First, we studied the cortical encoding of phonetic categories in infants with direct neural measurements based on EEG...”

And in the final paragraph (pg.15):

“In summary, this study demonstrated the emergence of phonetic encoding from 7 months of age using direct neural measurements during natural speech listening.”

I am not sure what "directly" means here – surely the authors do not wish to claim that these measures of neural responses constitute the way in which brains encode anything? That would be tantamount to claiming that the relation between neural physiology and cognition has been substantially solved. Given that EEG measures only a summed voltage across suitably aligned cortical (and subcortical) cells, the claim that these measures of scalp voltages are anything close to direct measures of cortical encodings cannot be taken seriously.

2. As an example of how small methodology details in the analytic pipeline from data to interpretation can change the interpretation, consider Figure 2B (pg.10) (I am assuming that the authors mean "Black circles indicate significance..." when they say "Black bars indicate significance..."). By changing the analysis window from 0-400ms to 100-500ms, 7-mo-olds go from not representing phonetic features to representing phonetic features. Could there be other parametric changes in this pipeline, or, more generally, other possible measures and even other methodologies that could reveal significant phonetic processing at 4 months? We cannot know because there are a vast number of such measurements and analytic parameters one could in principle make.

To put it differently, it seems too extreme to claim that these highly specific measures derived from scalp voltages are THE neural encoding of phonemes. From an ecological perspective too, it makes sense to value behavioral data over such neural data - for example, it is well known that while neurons in some secondary auditory cortical areas in adults are show invariant phoneme representations, others differ in their responses based on speaker voice characteristics. It is only in the context of a specific (ecological) task that the relevance of one or the other becomes apparent.

Therefore, the authors claim (Discussion section, pg.13): "...our results suggest that 4mo pre-babbling infants, despite being equipped with the fundamental combinatorial code for speech analysis, do not yet exhibit categorical phonetic encoding." is not supported.

3. The analysis windows mentioned in #2 are also a source of concern. The authors choose the time windows starting from the assumption (pg. 18): "A single input event at time t_0 affects the neural signals for a certain time window $[t_1, t_1+t_{win}]$, with $t_1 \geq 0$ and $t_{win} > 0$."

This assumption seems not well justified given (a) the extensive literature on predictions in the brain and (b) the regular presentation of "typical English language nursery rhymes" in this study – presumably well known to the adult participants.

That is, it is plausible that there are neural responses both to the predictions of phoneme sequences and to the acoustic perception of these phonemes; the former would presumably be a function of familiarity with the stimuli, and therefore, plausibly, a function of age or at least something that differs between infants and adults. I can imagine that disentangling these might be difficult.

4. I am not clear about the phonetic analysis that forms the basis for the TRF modeling. If I understand correctly, the spoken rhymes were converted into phoneme sequences, but it is not clear if, subsequently, the phonetic features were derived directly from the recordings or were essentially "mapped" on in a 1-to-1 manner; that is, if a given phoneme, say /b/, was always and automatically marked with a set of features like [+labial].

If the phonetic features were just such maps, then that authors are assuming that all occurrences of the phoneme /b/ in stressed (because they differentiate stressed/unstressed syllables) positions are all the same. Not only is this a very phonemic view of the stimuli, but it also leaves out allophonic and other idiosyncratic variations in the input. In fact, given the authors goals of looking for invariant "representations", I don't understand the design choice of including only a single speaker across all the recordings.

I am therefore not sure why the authors then need to insist that they are studying phonetic feature representations and not the development of phoneme representations (pg. 18/19), when there is already evidence for such invariant representations in adults. The authors indicate the link between overt phoneme processing and reading, but I don't see the relevance of those observations to the linguistic descriptions of the supposed internal language system which appears to require something like phonemes, at least primarily in the generative schools of linguistics.

5. I am not sure why the authors chose limited categories of features for their clustering algorithms (pg. 19) - only 3 for place and 3 for manner, and a third, 3-way distinction between vowel, voiced consonants and unvoiced consonants. I'm not even sure if the third is appropriate, given that there is behavioral and neural evidence for a distinction between vowels and consonants that extends beyond a merely sonority-based distinction.

This limits the kinds of inferences one can draw from the data. English phonemes rely on more nuanced distinctions, and behavioral data indicates that these are being tuned in the first year of life. Besides, there appear to be different timetables for the development of different phonetic features (again, from behavioral data). I can readily believe that their scalp voltage-derived measurements can show an increase in discriminability with age, but I'm not sure how these inform about the development of different phonetic features.

Such considerations might better help understand the Manner data in Fig. 2 (pg.12). In particular, the authors say (pg.10) that "phonetic feature encoding increased with age for place of articulation and voicing, but not manner of articulation..." However, from Fig.2B it appears that this is because of a low F-score (ie., low discriminability) in the adult sample for Manner - if we look at just the infant data, there seems to be an increase between 4-mo and 7-mo-olds, and no difference between 7-mo-old and 11-mo-olds.

I'm not sure what is a reasonable explanation for this pattern. In particular, why are adult F-scores for Manner the lowest of all age groups, when they are the highest for Place and Voicing? That is, while adult measures show the best discriminability scores for place and voicing, they show the worst discriminability scores for Manner. The authors provide no explanation for this.

We thank the reviewers and the editor for their work on this manuscript. The comments on the methodology and presentation of the results pushed us to further reflect on the procedure, leading to a substantial revision of the assessment metrics with an impact beyond this particular manuscript. The new results are based on a simpler methodology (e.g., we only use one lag window now, the MCCA analysis was excluded) and exhibit much stronger effects. In our revised manuscript we have also sought to address the concerns on novelty, ensuring that the key studies in this domain were referenced and revising our interpretation according to the reflections proposed by the reviewers.

We are grateful to the reviewers for their efforts in critiquing our work, which we believe have helped us to greatly strengthen our manuscript. We hope that you will receive this detailed revision positively and reconsider the manuscript for publication in *Nature Communications*.

Reviewers' comments:

Reviewer #1 (Remarks to the Author):

Understanding how infants process the features unique to speech and identifying in what ways and when infants' brain responses to speech resemble those of adults is one of the deep challenges in developmental cognitive neuroscience. The Liberto et al. manuscript reports a substantial advance in what we know about infant speech perception with this set of results. They use a novel method that combines cortical tracking of speech in infancy to investigate the emergence of phonetic feature encoding, and they use natural speech (nursery rhymes), which has never been done before. Whole-head EEG was used to examine cortical tracking, and the longitudinal design tested infants at 4, 7, and 11 months, as well as adults. The statistical analysis of the multivariate data produced by temporal response function (TRF) analysis is sophisticated. The results reveal a beautiful progression from 4-11 months, with ever-increasing precision of the neural encoding of phonetic features in speech. Importantly, non-speech control stimuli did not produce this change over time in the same babies. Phonetic feature categories are reflected in the neural data at both 7 and 11 months of age, but not at 4 months of age, and also reveal that at 11 months, infants have not yet reached adult neural processing (an important point). This is a benchmark study that will be a springboard for further work both on typically developing and at risk infant populations. It reaches, in my view, the level of innovation required for *Nature Communications*.

Reviewer #2 (Remarks to the Author):

The authors apply an analytical EEG approach by the first author to a longitudinal EEG data set of 4 mo, 7 mo and 11 mo old infants.

The authors claim that these data demonstrate for the first time how phonetic encoding emerges cortically, and that it does emerge or has emerged at 7 months of age. The study is rich in detail, but offers, at least in its current form, a string of somewhat frustrating analytical choices or shortcomings that leave open how justified some of the more bold conclusions

actually are. A major concern in my view is that phonetic encoding emerging in the first year of life has been amply demonstrated in arguably more compelling ways before.

We thank the reviewer for the insightful comments. While we answer the point on the novelty in the point-by-point reply below, we address the reviewer's concern on the analytical choices here.

It is true that the previous manuscript presented numerous methodological choices which were guided (or even anticipated) by previous work published in the last 8-10 years on various tasks (e.g., listening to monologues vs. selective attention), sound inputs (e.g., speech and music; audio vs. audio-visual), and cohorts (e.g., children, adults native speakers and second language learners, hearing-impaired older adults). For example, choices such as the time-lag window size and filters were based on Di Liberto et al., *Curr Bio*, 2015 and Di Liberto et al., *NeuroImage*, 2021. Rather than attempting to explain all those choices in detail, which would lead to a long and difficult paper without probably fully solving the reviewer's concern, we decided to revisit the analysis approach. This led to a much simpler solution involving less analytical choices that produced even stronger results, without the need for testing multiple lag-windows (a single larger lag-window is used now) and without the need for the MCCA analysis, which was excluded. The manuscript has been substantially revised to include these changes. Please find the details of the changes in the reply below.

I have organised my further comments below by perceived severity:

-- I. 69: Claims of primacy are usually problematic: I would simply not agree with this study being the "first to address the research questions directly"? Yes, the linear model obviously offers the advantage of tackling continuous speech, as the first author and many, many authors have demonstrated before, also in infants. This is a technical question, however. As for the two other question, "how are speech sounds encoded" and how speech sound representation develops over the first year of life, psycholinguist and phonological work has offered tremendous insight over the last decades.

As such, I see primarily an advance in technical fidelity here, by applying the Di Liberto 2015 analytical framework (in short, 'what do phonetic features add to an acoustic model of cortical speech encoding?') to infant data.

We are grateful that this point was raised as it pushed us to substantially improve the manuscript, which is now much clearer on what exactly is novel in this study, and how this can guide future research. The reviewer mentions two aspects that we would like to discuss separately in this reply: the methodological novelty and the novel insights into speech development.

Regarding the **methodological novelty**, the reviewer indicates that the advance here is primarily in technical fidelity, as many authors had already demonstrated the advantage of using continuous speech to study speech development. The manuscript might have been misleading in this regard. In fact, we certainly agree on the point that continuous speech was already used in infants research (including our own previous work). However, this is **the first time** that anyone has measured the **neural tracking of phonetic features** during continuous speech listening **in infants**. In fact, previous work either focussed on behavioural

paradigm (i.e., not using measurements of cortical electrical activity) or, when they used neural measurements, they focused on individual phonetic contrasts with, for example, oddball paradigms that study sound discrimination instead of continuous speech encoding. Even the Scientific report paper from the Iverson lab (mentioned in a comment that follows), which presented very interesting results, was limited in that sense, as it was constrained to vowel perception and because the stimulus was a continuous stream of vowels, which is very different from continuous speech. Neurophysiology studies with continuous speech focused on the overall acoustic responses (e.g., envelope tracking in Kalashnikova et al., Sci Rep, 2018 and Jessen et al., Dev Cog Neurosci, 2021), without isolating linguistic processing from acoustic processing. For these reasons, the neural indices isolated in the current study gave us an unprecedented view into how the human brain processes continuous speech. On top of that, the present study involved a longitudinal investigation of 50 babies instead of cross-sectional study of 12 babies (Kalashnikova) or 10 babies (Jessen), increasing the reliability and representativeness of the findings. In sum, in addition to the technological advance (i.e., the ability to measure the neural encoding of phonetic features with EEG in the first year of life), our results represent a substantial addition to the past literature as they indicate the exact trajectory for the neural encoding of phonetic feature categories in the first year of life, which no-one has previously measured longitudinally with neurophysiology and continuous speech sounds. We are not aware of any other work that could get such measurements and we did our best to clarify this aspect in the manuscript.

Regarding the **novel insights into how speech is processed in the infant brain**. We agree with the reviewer on that previous “psycholinguist and phonological work has offered tremendous insight” and we enhanced the references to that literature in the manuscript to clarify that. However, such a literature primarily focussed on syllables discrimination, **making such previous claims indirect** in that **they could not measure whether the human cortex was, in fact, encoding those distinctions as acoustically invariant, or whether simple acoustic features could explain the ability to discriminate those sounds**. Furthermore, most such studies relied on discrete paradigms involving discrimination of isolated syllables, with findings that may not apply to more complex continuous tasks such as nursery rhyme listening. We agree with the reviewer that the paper needed more clarity on that rich literature, which was indeed instrumental to our current understanding of speech perception in babies. This is exactly one of our points, as the present finding with nursery rhymes listening and EEG measurements is certainly in line with prior work (e.g., the work mentioned in P. Kuhl’s review in 2008) while suggesting that other prior work showing behavioural and MMR discrimination at 4mo probably have an acoustic basis rather than phonetic feature encoding. Finally, it was unclear before this study whether such previous results would have generalised to realistic nursery rhyme listening and, if so, whether the neural responses to phonetic features would have been sufficiently robust to be measurable with EEG.

These comments have been addressed in the text (e.g., paragraphs 1 and 2 of the introduction, and discussion on lines 248, 261, 323, and 332).

-- The arguably most important results (and the results most likely to motivate a large-scale, high-impact publication) are of borderline conventional significance: In lines 169f. The authors argue for phonetic–featural information improving model fits/predictive accuracy ins

11-mo and 7-mo olds (but not in 4-mo olds) with FDR-corrected p values all in the [0.023;0.05] range.

This is a very important point that we are glad was brought up, as this was actually a point of strength of this study, rather than a weakness. Our previous statistical analysis relied on the most conservative and heavily penalised approaches as possible. For instance, EEG prediction correlations are calculated by averaging results across all EEG channels (even channels that were not responsive to speech). Naturally the statistical results would show higher significance if our analyses had focused just on the most responsive electrodes. Nevertheless, we agree with the reviewer that the data in the first submission presented large variability across subjects (which is typical in infants EEG recordings), hampering the strength of the statistical results. To counter that challenge, we devised an improved assessment metric that we think is much more convincing. We thank the reviewer for their comment, which pushed us to devise this new solution, which we think others may find useful for TRF analyses in general.

The rationale is that the large variability across subjects is primarily due to the large differences in EEG noise between recordings. This is a well-known issue with EEG recordings, which is exacerbated by the limited single-subject data and by the noisy nature of the recordings with infants. In other words, the EEG prediction correlations vary a lot between subjects because EEG predictions for different subjects are correlated with different EEG signals with various levels of SNR while, ideally, EEG predictions should be correlated with the ground-truth neural signal hidden behind the EEG noise. Since that signal is not available, we estimated it by averaging the EEG signals across all subjects and channels. The resulting ground-truth estimate (the same for all subjects) was correlated with the EEG predictions, leading to EEG prediction correlations whose fluctuation would only be due to differences in the TRF models. The new results with this metric show patterns that are consistent with the previous submission, with much stronger statistical results (e.g., repeated measures ANOVA, $p=0.0009$ in Figure 2B, delta-band, showing that the EEG encoding of phonetic categories increases with age), that also allowed the observation of single-subject trajectories (Figure 1C). We think that this substantial revision addresses the major concerns of the reviewer.

-- Furthermore, the n.s. results for 4-mo olds are not given, but to my reading a formal comparison that would show that these results in 7 mo and 11 mo are indeed superior to those in 4 mo olds is missing. This would constitute an inference fallacy but might well be (and would need to be) rectified in revision.

That information has been added to the Result section in the revised manuscript and as part of the Supplementary Table 1.

-- In line 185f., in the very laudable canonical correlation analysis the authors turn to, it is notable that only the restricted time window yields a (borderline) significant effect, while the time window having yielded all previous effects more somewhat more substantially is far from significant.

Here again, a look at the TRF would have been helpful to understand the underlying response rather than the overly condensed predictive-accuracy measure only.

Also here, the so pivotal age-group comparison remains very cursory and qualitative: We are assured of significant effects at "large clusters of electrodes" from "7 months on", but again a more formal and stringent comparison of topographies by age groups is missing. It can thus not be known how meaningful the change in significant channels is.

This particularly relates/leads to my other main concern outlined, on the lurking/unknown role of signal/noise ratio in this analysis and conclusion.

In this revised manuscript, we opted for removing the MCCA analysis, a powerful analysis that, however, comes with additional analytical choices that complicate the procedure and explanation. Similar to MCCA, our new approach estimates a “ground-truth” denoised EEG signal, this time via a much simpler averaging procedure rather than a component analysis. In doing so, the resulting analysis produces robust results that are summarising the overall encoding across all EEG channels. Regarding the comment on possible “topographical changes”, it is important to note the substantial differences in head anatomy across the first year of life, which likely lead to large variability both within and between participants. For that reason, we did not explicitly test for topographical differences, as we don’t think that result would have been particularly informative in this particular case.

-- p. 11/Discussion: As stated in other terms in other instances of my review, the conclusion offered (“first evidence that the human cortex encoded phonetic categories during the first year of life”) is overly strong: There can hardly be any doubt that the work by Werker, Kuhl, Näätänen and others (major uncited paper eg Stager & Werker, Nature 1997) has demonstrated a striking change in phonetic category perception between approx. 4 and approx. 11 months of age. I applaud the approach to use the di Liberto “phonetic-features plus acoustics” modelling approach to infant data, but the new insight appears to be mainly of technical nature. Moreover, the data themselves leave worries over how robust the claimed “gradual” change in predictive accuracy, TRF change, and topography really is. The authors go at great length in the discussion justifying the novelty of their findings. They argue in particular that the present study is first to elucidate the precise time point. For reasons developed in my more technical comments, I am not sure that the data manages to demonstrate such a “pivot point” between 4 and 7 months. I do think that a previous uncited study also from UCL (Iverson lab, Scientific Reports, 2017 <https://www.nature.com/articles/s41598-019-55085-y>) has done this, using what the present authors call “sound discrimination metrics”, somewhat more compellingly (as they tackle the emergence of the ‘vowel space’ quite directly and with compelling age group x vowel-pair interactions). I am just giving this example in somewhat more detail to illustrate how the current paper overemphasises novelty in places where the analyses potentially don’t live up to these claims.

We thank the reviewer for the very useful references. We have revised the text to more precisely explain the novelty, including a more exhaustive discussion on how that pushes forward previous work. In light of that, we agree that some aspects of the paper had to be clarified or toned down, which we did throughout the manuscript. Nevertheless, we think that this study is much more than a technical contribution. Even by considering the additional references suggested by the reviewer, previous work focussed on syllable discrimination (e.g., with meaningless syllables or monosyllabic words and nonwords) and/or on limited phonetic contrasts.

Stager & Werker, Nature 1997 relied on a clever behavioural paradigm (based on switch trials and looking time) to study speech sound discrimination and word learning in 8 and 14 months old babies. That (and other) studies could show that there is a change in phonetic

category perception in early development. However, that perceptual change could have been underpinned by a variety of changes in the neural encoding of speech, and in fact that particular paper does not contain any particular claim on neural implementation or encoding. Instead, our EEG results allowed us to test the hypothesis that an acoustically-invariant neural encoding of phonetic feature categories emerges during the first year of life.

Regarding the paper from the Iverson lab, Scientific Report, 2017, while it is interesting, it only studies auditory responses to seven vowels with stimuli consisting of continuous sequences of vowels, which is far from real speech. Here, one element of novelty is the simultaneous investigation of 27 phonemes (vowels and consonants), which was not done before. Second, previous study clarified that they were focusing on acoustic responses that are likely to originate in primary auditory cortex (which is a problem for paradigms using discrete syllables, as the EEG response is dominated by the acoustic onsets, differently from continuous speech), while the phonetic encoding measured here (the EEG prediction gain is more likely originating from STG (e.g., see the review from Eddie Chang's team, Neuron, 2019 <https://www.sciencedirect.com/science/article/pii/S0896627319303800>; note that there is also a study in preparation, which was presented at the ARO conference, on intracranial recordings showing phonetic feature encoding in STG but not in primary auditory cortex: Raghavan et al., and Mesgarani, in preparation). For these (and other) reasons, we think that our results are more than a substantial methodological advance.

-- Lastly, I agree with the authors that a difference in SNR is unlikely to explain all effects, as the acoustic-envelope impact seems not too much affected by age group. However, there is a limit to this argument: As in most models using the di Liberto approach (incremental models of acoustic plus phonetic features), the phonetic features can only add explaining a certain, relatively minor remaining part of the variance. If there were a systematically lowered SNR in the younger EEG data, I would expect the relatively most impactful regressors (acoustic) to work fine nevertheless but more subtle effects to become invisible first (such as phonetics). In short, a more explicit demonstration that SNR is not a worry at all when arriving at this paper's conclusions would be in order.

We conducted additional analyses to confirm that the results could not be explained by differences in SNR between groups. Note that, if anything, we would have expected stronger SNR in younger babies because of the skull thickening with age in the first year of life. Of course, there are other important factors to consider, such as movement, which can change a lot between different age groups. To address this issue quantitatively, we measured SNR by comparing ERPs to the word acoustic onsets with a pre-stim baseline. There was no difference in SNR across conditions. This result was added to the revised manuscript (from line 162).

Figure caption: SNR was calculated as the ratio between post- and pre-stim power for the ERP calculated for the first word in each trial. A one-way ANOVA indicated no significant effect of SNR ($F(2,98)=0.9$, $p=0.4013$). If anything, the non-significant trend is opposite to what is found for the phonetic feature encoding.

MINOR:

-- Related to major points above, the hypothesis (spelled out in line 98f) seems to be that “neural encoding of phonetic features during natural speech listening is already developing during the first-year of life”. This strikes me as an odd place to start, given that in sentence 1 of the abstract we already learn of this very fact. See other point on claimed novelty.

The abstract is referring to the skill of recognising speech sounds such as words. Our hypothesis (which is supported by the data) was that such a skill is underpinned by a neural encoding of phonetic categories. That was not obvious, as we can recognise sounds without the need for categorical perception of phonetic features. The introduction clarified this distinction on lines 108 and 113.

-- l. 130 and many other instances: p values are not a measure of effect size. Please also report meaningful effect sizes, here, average r_{predict} . If in-text the information gets too dense, I consider the r more informative than the p value.

The new figures report the data with violin plots (Fig.1B and 2B) as well as single-subject trajectories (Fig.1C). We think that this is a clear and informative way of reporting the results, and reporting the mean r -values for every plot would make the text too dense. Indeed, we would also be happy to share the exact numerical results in the supplementary materials, if the reviewer thinks it's necessary, but we think the new violin plots are sufficiently clear in that regard.

-- Please allow the reader to judge the TRF model/transfer function to be seen. Previous research tells us of substantial differences in infant ERPs, and accordingly also in infant TRFs. TRFs (including informative baseline/negative time lags) should be included for all age groups, as any model prediction as a whole obviously depends on the set of weights (ie, the TRF) creating such a prediction.

(I later saw that Figure 2 contains some featurally resolved TRFs in image plots; I here referred to more conventional, ERP-like temporal response functions for the acoustics.

What Figure 2 resolves, are “phonetic–temporal receptive functions” of sorts, no?)

If we understand correctly, the reviewer is asking to report ERP-like TRFs in response to the acoustics, such as the more typically used envelope TRF. However, we did not use

envelope TRF models here, so such a model would not reflect any of the other results. We could indeed average the TRF weights across different features to get a TRF that is visually similar to an ERP. For example, averaging the weights for the spectrogram TRF would give something similar to an envelope TRF. However, we think it is more appropriate to show the weights of the actual models that led to the results in Figures 1 and 2.

While Figure 2C presents only the weights of interest for simplicity (only S and F, while the nuisance regressors were excluded; only 0-400, as the rest was removed due to possible side artifacts), we also included all the model weights in Supplementary Figure 1.

-- The lack of individual data in Figure 1B and in Figures 2–3 (see note on Figure numbering) is not state of the art. Please allow your readers an impression of the actual variability in these data rather than only the SEM. (Note that I infer the error metric here; the figure caption lacks details).

This is especially warranted as these are longitudinal/dependent data! A great deal of information is absent at least from the figures by not emphasising/displaying the longitudinal trajectories of individuals infants (plus the aggregate/average, of course).

The revised paper includes the distributions and single-subject result/trajectories where appropriate.

-- A misnumbering of figures has happened, the label Figure 2 is used twice.

Fixed

-- Open science/open data: I was missing a statement on how the authors plan to make these data and models publically available?

Please see the Data and Code Availability section from line 459.

Reviewer #3 (Remarks to the Author):

SUMMARY

The authors investigate phonetic feature encoding in infants (ages 4, 7 and 11 months) using EEG, and compare this to adult data. They test the hypothesis that phonetic encoding emerges with development, by comparing the unique explained variance of a phonetic feature model above a spectrogram/envelope acoustic model. They find that, indeed, a phonetic model explains additional variance as a function of age, and that the organisation of feature responses (relative dissimilarity) also becomes more adult-like. The authors conclude that phonetic feature encoding emerges during the first year of life.

Overall, I commend the authors on a true data collection tour de force, in order to answer an interesting research question. However, there are a number of experimental decisions that were omitted or not fully motivated in the main manuscript, and some overstepping of claims, which I would need to see revised before recommending this manuscript for publication.

We thank the reviewer for the positive feedback. The revised manuscript presents a substantially improved analysis pipeline with simpler models and less analytical choices that are now fully motivated in the main manuscript. We also clarified the hypotheses and carefully adjusted the terminology and claims, making the manuscript much clearer. Please find our point-by-point reply below.

MAJOR

[1] Misleading description of stimulus and task in main text

In the main text of the manuscript, the authors emphasise that they are investigating responses to “continuous natural speech”; however, it was not until I read the methods that I realised they actually used (i) audio-visual materials with (ii) sung speech. This was very hidden away, almost misleadingly so. Of course the accompaniment of visual speech cues and the use of an extremely prosodic stimulus will have a big impact on speech encoding, but this is not properly addressed in the manuscript, other than a small comment in the discussion.

To address this, I believe that the authors need to:

- make it clear in the main text (introduction) that they are using audio-visual sung speech
- provide an explicit analysis to show that this is representative of speech processing without visual accompaniment
- show that this is representative of true natural, non-sung, speech
- provide a demonstration that the gain in their model performance is not due to the more robust integration of audio and visual information with age

We agree with most of the comments and revised the manuscript accordingly. First, the revised text is much clearer regarding the particular stimulus used in the experiment and the implications for speech perception and processing (e.g., lines 60, 81, 88, 249, and 310). Next, we would like to comment on the point of the visual input. The visual stimulus was included for making the experiment more ecologically valid and engaging for the infants, as the infant literature shows that learning phonetic categories in infancy is a social and multi-modal process (e.g., Kuhl, 2007). Infants learn language best in situations of direct eye-to-eye contact when they are “online for learning” (Csibra, 2010), and behavioural data suggest that little of value regarding phonetic categories is learned by “overhearing” speech (i.e. auditory only input, Kuhl et al., 2003). So to have an ecologically-valid task with infants, it is best to have an audio-visual stimulus. Multisensory audio-visual effects have been previously measured with natural speech listening paradigms and TRF analyses in adults. However, those effects are typically very small and difficult to measure and, in fact, are usually studied in challenging listening scenarios (e.g., speech-in-noise) to maximise the role of visual input (e.g., Crosse, Di Liberto, Lalor, J. of Neuroscience, 2016).

This study also included two strategies for more directly assessing the impact of the visual input. The first strategy consisted of including a nuisance regressor describing the overall visual motion in the multivariate TRF analysis (Figure 2). Removing that regressor did not change the result. In light of this, while the visual input would be expected to have a role in phonetic category learning, our evaluation suggests that it did not exert any systematic effects on our EEG results. These considerations have been added to the discussion section (see lines 317). Furthermore, we also collected eye-tracking measurements to quantify the total

time each participant looked at the screen and did not find significant differences across the three infant groups. Infants spent comparable amounts of time watching the screen during the nursery rhyme presentation at 4-, 7- and 11-months of age (mean = 22.0%, SD = 19.2%; mean = 38.5%, SD = 15.9%; and mean 37.5%, SD = 17.6% respectively). However, please note that the Tobii eye-tracker isn't reliable at detecting the infant pupil below 6- months of age (as indicated by the manufacturer), which caused the eye-tracking traces to contain segments with no data at all. Since these measurements were unreliable, we preferred not to report them in the manuscript and carry on further analyses and considered the other control analysis sufficient to our goals.

[2] Signal to noise ratio as an alternative explanation of results

I feel that an obvious counter explanation of the present results is that the signal to noise ratio of the datasets will differ at different ages. I imagine that the data quality improves as a function of age, with knock-on consequences for the modelling results, but I did not see this issue addressed in the manuscript. Furthermore, as infants develop, their sensory systems improve with age, also. So from the signal perspective, it is likely that the older infants' auditory cortices receive a more robust representation of the stimulus input than the younger infants.

This issue is exacerbated by the lack of a baseline stimulus or baseline analysis. I understand that the main analysis is a subtraction of model fits, of the phonetic+acoustic model above the acoustic model, but it would be helpful to see the model performance curves of each model going into that subtraction, and well as basic visualisation of the resulting TRF kernels.

Furthermore, it is possible that a less noisy neural signal naturally favours the more sparse feature model over the complex spectrogram model. It would be helpful to test this empirically, for example by simulating results under different noise conditions.

Regarding the simulation analysis, similar analyses were conducted in recent previous studies that assessed how forward models are affected by signal-to-noise ratio (SNR; Crosse, Zuk, **Di Liberto** et al., *Frontiers in Neuroscience*, 2021) and temporal imprecision (Carta and **Di Liberto**, *Journal of Neuroscience Methods*, 2022). While we have not conducted a simulation on the specific set of stimuli in this experiment, that previous work clarified how TRF models would "behave" in general in relation to changes in SNR and temporal imprecision. For example, what if phonetic features were represented but with an imprecise timing? Based on Carta and Di Liberto 2022, EEG prediction correlations would be more robust to such imprecisions than TRF weights, motivating our stronger reliance on prediction metrics in this study.

Regarding the experimental side, we expected that skull thickening with age would have been a major contributor to a possible change in SNR. Furthermore, 7 and 11mo typically move more than 4mo infants. So, if anything, we expected a reduction in SNR with age. To verify whether SNR could contribute to the key results of this study, we conducted an explicit analysis where we quantified SNR as the ratio of the power of the post- and pre-stimulus

onset ERP, calculated for the first word of each trial, and found that SNR could not explain the change in phonetic feature encoding with age.

Figure caption: SNR was calculated as the ratio between post- and pre-stim power for the ERP calculated for the first word in each trial. A one-way ANOVA indicated no significant effect of SNR ($F(2,98)=0.9$, $p=0.4013$). If anything, the non-significant trend is opposite to what is found for the phonetic feature encoding.

[3] Research questions

[a] In a number of places in the introduction, and again in the statement of 3 research questions (pg 3), the authors allude to investigating how speech sounds are “perceived” by infants. For example, research question 1 is “How do infants perceive and encode...” — the task is passive listening, and so I ask the authors to be careful not to describe questions regarding how neural encoding relates to the eventual perception.

We revised that terminology throughout the manuscript (e.g., line 76, 86), as this study investigates the *encoding* of speech sounds.

[b] Furthermore, if we remove the “perception” part of the first research question, the first two questions become extremely redundant: “how infants encode phonological units” and “how are these speech sounds encoded” seem the same to me.

That sentence was stating three open questions, so that was not changed. Instead, we revised the following sentence, where we explain that this study investigates the neural “encoding” of speech sounds.

[c] In the final paragraph of the introduction, the authors restate their question as “Test the hypothesis that the neural encoding of phonetic features during natural speech listening is already developing during the first year of life” — this is a vague hypothesis, and the way it is stated, I am not sure that the outcome could be any other way.

That sentence was rephrased. Our hypothesis was that the EEG encoding of phonological categories would have shown a progressive increase with age.

Overall, I ask the authors to be more pinpointed in their research questions, and the true hypothesis space that they are investigating.

[4] Analysis decisions lacking motivation

[a] Why was 1-15 Hz chosen for the analysis?

Good point. That choice was based on previous EEG work on adults and narrative speech listening (Di Liberto et al., *Current Biology*, 2015). The revised version focuses, instead, on 0.1-8 Hz, which is more relevant to this specific experiment. The high-pass filter is set at 0.1Hz for denoising reasons. The low-pass filter is set to 8Hz as a previous study on this dataset (Fig.1 from Attaheri et al., *NeuroImage*, 2022) showed that most of the signal power is below that frequency cut-off. The motivation is not indicated on lines 392 and 394.

[b] What is the motivation for separating stressed and unstressed syllables? Wouldn't a true phonetic model be invariant to that distinction?

The initial rationale was that infants would more easily and robustly encode phonological information of the stressed syllables, leading to responses with different latencies and strength for stressed and unstressed syllables. However, the reviewer's comments made us realise that it was an overcomplicated analysis at this stage. As such, we re-run the analysis with a simpler model where stressed and unstressed information was not separated. While the results are overall similar to the previous submission, the underlying procedure is much simpler, hence preferable.

[5] Window selection

The time-window of 0-400 ms is extremely long considering that the phoneme duration is only about 100 ms, and the latency of phonetic encoding is typically around 100 ms. What long-latency responses are the authors expecting to capture with their encoding models? As the window increases in duration, the authors will be capturing more and more statistical structure of the neighbouring sound sequences, which seems problematic for the current research questions if the aim is to test instantaneous phonetic encoding.

Further to this point, the authors also provide an analysis using 100-500 ms, but there is no explicit comparison of whether the additional 100 ms provides explanatory power. It seems to me that the proper way of testing whether responses are longer lived in infants would be to see if a 0-500 ms model delay provides unique variance above the 0-400 ms model; if not, then do not use a longer window; if it does, then only use the longer window.

The TRF analysis is particularly suited for dealing with the exact issue that the reviewer mentioned. In fact, ideally, the window should be longer rather than shorter, as a long window would allow us to more clearly distinguish time-latencies that are more or less important for the stimulus-EEG relationship studied (for example, consider that other TRF/deconvolution approaches extract the maximum possible window, such as Ross Maddox's approach in Shan, Cappelloni, and Maddox, *bioRxiv*, 2022 and other previous studies). Another thing to keep in mind is that all participants listened to the same stimuli (i.e., with the same temporal regularities).

We realise that the methodological choice of using two windows was not sufficiently explained and it appeared overcomplicated. One reason was that we could not really know a priori what the latency of a phonetic-feature TRF should be in infants. As such, we preferred to select a larger lag-window. In this revised submission we present an analysis with a window from -100ms to 500 ms. The reason for including the negative lags is purely for visualisation and interpretation (see lines 142-143). The mTRF produces "side artifacts"

(similar to filtering artifacts) that make it difficult to interpret the results for latencies at the sides of the lag window. We think that the overall narrative has improved as a result.

MINOR

Speech TRFs do not necessarily reflect neural entrainment (pg4, line83) — they can also just reflect instantaneous evoked responses. Please remove

The revised manuscript clarifies (line 93) that we refer to neural entrainment in the broad sense (which can also reflect evoked responses), as defined by Kayser and Obleser, 2019.

Over-use of bar plots — should show the underlying distribution of the data

We use violin plots now and show individual subject trajectories in Fig. 1C.

Adults were recorded with a higher density cap, and I imagine that their spline procedure reduces density while retaining signal. This brings the concern that the adult data is inherently of a higher SNR, simply because of the difference in hardware.

Interesting reflection. The adult data is used as a reference really, and it's not compared directly with the infants data, as the hypothesis was only on the infants longitudinal data. So, in any case, we don't think this would be an issue even if that was the case. While we may agree in principle, mapping 128 to 64 channels would not change the results in practice (but that may be the case if we were going to 16 or 8 electrodes). A spline interpolation would change the noise, in the sense that, a spatially localised noise (a bad electrode) would spread across all other channels, so it would become smaller, but would impact more channels. So, it's debatable whether a forward TRF model, where EEG prediction correlations were averaged across all electrodes, would actually benefit from such a procedure. If the reviewer thinks it's appropriate, we will re-run the analysis by instead considering a selection of 64 channels (without spline interpolation). However, we don't expect to see any differences and, even if that was the case, that would not impact our results and interpretations.

Reviewer #4 (Remarks to the Author):

This paper uses multichannel EEG coupled with multivariate Temporal Response function(TRF) modeling to examine brain-derived responses to phonemes in speech in 4-, 7-, & 11-mo-olds, and in adults.

While I generally like the research direction and the experimental work, the motivation for the study is somewhat bombastic given the state of neuroscience today. Besides, there are a few, somewhat important methodological considerations that call the results into question. I recommend a revision, and detail my concerns below.

We thank the reviewer for the positive comments. The major comments have been addressed in two main ways: First, the text has been revised according to the reviewer's comments. In particular, the motivations, hypotheses, and interpretation have been revisited with more precise and consistent terminology throughout the manuscript. Second, the

analytical procedure has been substantially revised, leading to a much simpler set of analyses and fewer methodological choices.

1. The authors state early on (pg. 3 [always referring to the PDF pages of the merged ms]):

“However, behavioural methods can only serve as an indirect index of the emergence of linguistic skills, and cannot reveal when the phonetic encoding in the human cortex becomes invariant across different instantiations.”

And, in the next paragraph,

“This study is the first to address these research questions directly.”

And reiterated in the Discussion section (pg.12):

“First, we studied the cortical encoding of phonetic categories in infants with direct neural measurements based on EEG...”

And in the final paragraph (pg.15):

“In summary, this study demonstrated the emergence of phonetic encoding from 7 months of age using direct neural measurements during natural speech listening.”

I am not sure what “directly” means here – surely the authors do not wish to claim that these measures of neural responses constitute the way in which brains encode anything? That would be tantamount to claiming that the relation between neural physiology and cognition has been substantially solved. Given that EEG measures only a summed voltage across suitably aligned cortical (and subcortical) cells, the claim that these measures of scalp voltages are anything close to direct measures of cortical encodings cannot be taken seriously.

The revised manuscript clarifies that with “directly” we were referring to the direct measurement of neural electrical activity, as opposed to indirect measurements such as fMRI (which measures changes in blood oxygen levels) and behavioural studies, which do not directly inform us on the neural encoding of speech. Indeed, EEG measurements are spatially coarse and represent the overall neural activity of various brain areas, and definitely not single neurons (instead, it’s more like hearing the crowd at a stadium, where no individual voice can really be identified, just more or less where the sound is stronger or weaker). Rather than the where (which neuron or cortical area) or the how (in which way the information is encoded exactly), EEG is capable of informing on “when” the human cortex is responding to a given stimulus. In this case, our TRF models are informing us on what speech-EEG latencies are most important to describe the relationship between the EEG signal and different spectrogram frequencies or phonetic features. And it turns out that the EEG responds with different temporal dynamics to different phonetic features. While that may not reflect the activity of any single neuron, it is indeed a direct electrical measurement that is resulting from cortical signals. So, even if spatially coarse and unspecific in terms of “how” the information is encoded, we think it is safe to say that the TRF reflects at least the temporal encoding (“when” the cortex responds to selected speech features).

On a separate note, and while we can’t claim anything on cortical sources in this study with EEG, previous studies with invasive EEG recordings gave us a precise idea of where the EEG prediction gain is likely to originate. Posterior STG seems to be the most likely area, and the encoding of phonetic features appears distributed. Note that that invasive EEG used high-gamma signals, which are related to spike rate. For example, Mesgarani et al., Science, 2014, showed the encoding of phonetic features in posterior STG (see also the review from Eddie Chang’s team, Neuron, 2019

<https://www.sciencedirect.com/science/article/pii/S0896627319303800>). This was also confirmed by another recent study that was presented at the ARO conference using intracranial recordings, which showed phonetic feature encoding in STG but not in primary auditory cortex (Raghavan et al., and Mesgarani, in preparation). Again, this is just to mention that we do have hypotheses on how (distributed encoding) and where (pSTG) phonetic features may be encoded. Indeed, low-frequency EEG is not the same as high-gamma invasive EEG, and we have revisited the manuscript to ensure that our motivations, hypotheses, and claims are clear.

2. As an example of how small methodology details in the analytic pipeline from data to interpretation can change the interpretation, consider Figure 2B (pg.10) (I am assuming that the authors mean “Black circles indicate significance...” when they say “Black bars indicate significance...”). By changing the analysis window from 0-400ms to 100-500ms, 7-mo-olds go from not representing phonetic features to representing phonetic features. Could there be other parametric changes in this pipeline, or, more generally, other possible measures and even other methodologies that could reveal significant phonetic processing at 4 months? We cannot know because there are a vast number of such measurements and analytic parameters one could in principle make.

The revised manuscript presents a substantially simplified analysis procedure with less and better motivated analytical choices and more robust results. We are now only using a single, large lag window (-100 to 500 ms) which includes the response latencies previously seen for adults with additional lags at the start and end of the window. Furthermore, the MCCA analysis (which was effective, but included other parameters such as the number of components to retain) was removed. Furthermore, stressed and unstressed syllables were considered together as in previous studies, reducing the overall number of features. Finally, we first conduct a simple analysis on the S and F features alone. Then, we present the analysis with the combined models, making the narrative more complete and linear.

We think that the new analysis, with much less analytical choices, simpler models, and cleaner narrative led to a substantial improvement of the manuscript. Thank you for highlighting the issue.

To put it differently, it seems too extreme to claim that these highly specific measures derived from scalp voltages are THE neural encoding of phonemes. From an ecological perspective too, it makes sense to value behavioral data over such neural data - for example, it is well known that while neurons in some secondary auditory cortical areas in adults are show invariant phoneme representations, others differ in their responses based on speaker voice characteristics. It is only in the context of a specific (ecological) task that the relevance of one or the other becomes apparent.

That is true and an interesting point. Behaviour is key, and it is not our intention to diminish the value of behavioural studies. As it was mentioned by Gomez-Marin and Ghazanfar (Neuron, 2019), Neuroscience needs behaviour. Nevertheless, behaviour is particularly challenging to study in infants, and we demonstrated that neurophysiology can be used to study ecologically-valid scenarios (such as nursery rhymes) that we don't know how to study behaviourally. We think there is great value in this and we think that the revised manuscript is much clearer on this point.

Therefore, the authors claim (Discussion section, pg.13): "...our results suggest that 4mo pre-babbling infants, despite being equipped with the fundamental combinatorial code for speech analysis, do not yet exhibit categorical phonetic encoding." is not supported.

This is a speculative claim based on our results, which are consistent with the rich literature on perceptual narrowing. We adjusted the text by being more specific about what we measured (EEG prediction gain for 7 and 11mo but not 4mo in delta-band), and changed "suggest" to "speculate" (from line 286).

3. The analysis windows mentioned in #2 are also a source of concern. The authors choose the time windows starting from the assumption (pg. 18): "A single input event at time t_0 affects the neural signals for a certain time window $[t_1, t_1+t_{\text{win}}]$, with $t_1 \geq 0$ and $t_{\text{win}} > 0$."

This assumption seems not well justified given (a) the extensive literature on predictions in the brain and (b) the regular presentation of "typical English language nursery rhymes" in this study – presumably well known to the adult participants.

That is, it is plausible that there are neural responses both to the predictions of phoneme sequences and to the acoustic perception of these phonemes; the former would presumably be a function of familiarity with the stimuli, and therefore, plausibly, a function of age or at least something that differs between infants and adults. I can imagine that disentangling these might be difficult.

There was a typo in that statement. We meant $t_1 \geq t_0$. That was fixed in the revised manuscript. Note that the statement represents the standard assumption for the widely used ERP analysis (same for TRF analyses).

The reviewer is referring to the possibility of having neural signals anticipating the stimulus, which is absolutely possible (e.g. Leonard et al., Nature Comm, 2016, a study with intracranial EEG). Nevertheless, to our knowledge, that was not seen in previous TRF studies on continuous speech nor music with non-invasive EEG nor MEG (e.g., Di Liberto et al., Curr Bio, 2015; Di Liberto et al., eLife, 2020). Instead, top-down predictions seem to impact the neural response after the stimulus (e.g., by modulating the acoustic response, rather than producing an evoked response before the stimulus, as in Broderick et al., 2020). In light of this, we did not expect anticipatory responses here. Nevertheless, we also included negative lags (for a different reason, as those negative lags help absorb the side artifacts of the TRF, facilitating visualisation of the positive lags).

4. I am not clear about the phonetic analysis that forms the basis for the TRF modeling. If I understand correctly, the spoken rhymes were converted into phoneme sequences, but it is not clear if, subsequently, the phonetic features were derived directly from the recordings or were essentially "mapped" on in a 1-to-1 manner; that is, if a given phoneme, say /b/, was always and automatically marked with a set of features like [+labial].

The latter is correct. We clarified that on line 429..

If the phonetic features were just such maps, then that authors are assuming that all occurrences of the phoneme /b/ in stressed (because they differentiate stressed/unstressed syllables) positions are all the same. Not only is this a very phonemic view of the stimuli, but it also leaves out allophonic and other idiosyncratic variations in the input. In fact, given

the authors goals of looking for invariant “representations”, I don’t understand the design choice of including only a single speaker across all the recordings.

I am therefore not sure why the authors then need to insist that they are studying phonetic feature representations and not the development of phoneme representations (pg. 18/19), when there is already evidence for such invariant representations in adults. The authors indicate the link between overt phoneme processing and reading, but I don’t see the relevant of those observations to the linguistic descriptions of the supposed internal language system which appears to require something like phonemes, at least primarily in the generative schools of linguistics.

These are very interesting points that we would love to study in the future, but it’s beyond the scope of this investigation. This was the first study of its kind in infants, so we chose an approach that we had assessed to be feasible with the amount of data involved. Future studies may increase the number of speakers and analytic considerations involving, for example, allophones.

The intuition here was that all occurrences of a phoneme (e.g., /b/) present different context and acoustics. We hypothesised (as done in previous studies) that if some level of invariance was present, then the EEG prediction gain would have reflected that. Of course, we are not claiming that this is the exact invariance encoded in the human cortex, as it is possible that a different level of detail is instead applied (e.g., allophones, or a different set of features). Now that we have shown that some level of invariance can be measured (it was not obvious that this would have emerged from the EEG signals), future studies may explore all sorts of questions exactly as the reviewer suggested. There is much more to do on this and we consider this study a first step into that line of work.

5. I am not sure why the authors chose limited categories of features for their clustering algorithms (pg. 19) - only 3 for place and 3 for manner, and a third, 3-way distinction between vowel, voiced consonants and unvoiced consonants. I’m not even sure if the third is appropriate, given that there is behavioral and neural evidence for a distinction between vowels and consonants that extends beyond a merely sonority-based distinction.

This limits the kinds of inferences one can draw from the data. English phonemes rely on more nuanced distinctions, and behavioral data indicates that these are being tuned in the first year of life. Besides, there appear to be different timetables for the development of different phonetic features (again, from behavioral data). I can readily believe that their scalp voltage-derived measurements can show an increase in discriminability with age, but I’m not sure how these inform about the development of different phonetic features.

Such considerations might better help understand the Manner data in Fig. 2 (pg.12). In particular, the authors say (pg.10) that “phonetic feature encoding increased with age for place of articulation and voicing, but not manner of articulation...” However, from Fig.2B it appears that this is because of a low F-score (ie., low discriminability) in the adult sample for Manner - if we look at just the infant data, there seems to be an increase between 4-mo and 7-mo-olds, and no difference between 7-mo-old and 11-mo-olds.

I'm not sure what is a reasonable explanation for this pattern. In particular, why are adult F-scores for Manner the lowest of all age groups, when they are the highest for Place and Voicing? That is, while adult measures show the best discriminability scores for place and voicing, they show the worst discriminability scores for Manner. The authors provide no explanation for this.

After careful consideration, we opted for removing the clustering analysis which was particularly sensitive to single subject variability in the TRF weights and could only lead to significant results by using resampling methods, differently from the other analyses (e.g., EEG prediction correlation) which were robust to the point that we thought relevant to show single subject trajectories (Fig. 1C).. A better denoising approach will have to be devised in the future for more reliably studying the phoneme maps on this kind of data.

REVIEWERS' COMMENTS

Reviewer #2 (Remarks to the Author):

Let me thank the authors for an exceptionally erudite, thoughtful, and creative reply to my concerns, and the according changes to the manuscript, which have most certainly led to a more convincing and more transparent overall presentation. I will certainly not stand in the way of these data being published as they are, and hope that readers will profit from our exchange (if reviews are being made public).

Jonas Obleser

Reviewer #3 (Remarks to the Author):

I thank the authors for taking my comments into account.

I would like to revisit my comment regarding how the research questions are stated: "1) How do infants process phonological units such as syllables and phonemes in continuous natural speech? 2) How are these speech sounds encoded in the infant brain? And 3) how does that encoding develop across the first year of life?"

I do not understand what "how" the authors are able to contribute with this work. It seems to me that the only question this study can answer is: "Does phonetic information become more robustly encoded across the first year of life?". The exact coding schemes and processing mechanisms are inaccessible with the analyses they conducted.

We are grateful to the reviewers for their constructive efforts in critiquing our work and for their endorsement. Please find below the point-by-point reply to the final comments, which is also reported in the author checklist document. Thank you again for this very positive and useful review process!

Reviewers' comments:

Reviewer #3 (Remarks to the Author):

I thank the authors for taking my comments into account.

I would like to revisit my comment regarding how the research questions are stated: "1) How do infants process phonological units such as syllables and phonemes in continuous natural speech? 2) How are these speech sounds encoded in the infant brain? And 3) how does that encoding develop across the first year of life?"

I do not understand what "how" the authors are able to contribute with this work. It seems to me that the only question this study can answer is: "Does phonetic information become more robustly encoded across the first year of life?". The exact coding schemes and processing mechanisms are inaccessible with the analyses they conducted.

We thank the reviewer for the constructive comments and for highlighting this issue with the statement of our objectives. We acknowledge the point, and we have changed our text accordingly (lines 63-66). Furthermore, we made that point clearer throughout the discussion.